

# Seven-dimensional super Yang-Mills at negative coupling

Joseph A. Minahan[1,2][⋆], Usman Naseer[1,†] and Charles Thull[1,‡]

**1** Department of Physics and Astronomy, Uppsala University,
Box 516, SE-751 20 Uppsala, Sweden
**2** Center for Theoretical Physics, Massachusetts Institute of Technology,
Cambridge, MA 02139, USA

⋆ joseph.minahan@physics.uu.se, † usman.naseer@physics.uu.se, ‡ charles.thull@physics.uu.se

## Abstract

We consider the partition function for Euclidean $SU(N)$ super Yang-Mills on a squashed seven-sphere. We show that the localization locus of the partition function has instanton membrane solutions wrapping the six "fixed" three-spheres on the $\mathbb{S}^7$. The ADHM variables of these instantons are fields living on the membrane world volume. We compute their contribution by localizing the resulting three-dimensional supersymmetric field theory. In the round-sphere limit the individual instanton contributions are singular, but the singularities cancel when adding the contributions of all six three-spheres. The full partition function on the $\mathbb{S}^7$ is well-defined even when the square of the effective Yang-Mills coupling is negative. We show for an $SU(2)$ gauge theory in this regime that the bare negative tension of the instanton membranes is canceled off by contributions from the instanton partition function, indicating the existence of tensionless membranes. We provide evidence that this phase is distinct from the usual weakly coupled super Yang-Mills and, in fact, is gravitational.

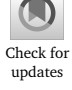

# 1 Introduction and summary

The gravity duals of nonconformal gauge theories can lead to new insights on both sides of the correspondence. The duals of maximally supersymmetric gauge theories in flat $p + 1$ dimensions were first found by taking the near horizon limit of a stack of D$p$ branes [1]. Except for $p = 3$, all such gauge theories are nonconformal. In order to make direct quantitative comparisons between the gauge theories and the supergravity duals, it is helpful to put the theory on a Euclidean sphere. The first direct check was made in [2], where the authors found a consistent truncation of five-dimensional gauged $\mathcal{N} = 8$ supergravity which correctly reproduced the free energy of $\mathcal{N} = 2^*$ $SU(N)$ gauge theory on $\mathbb{S}^4$ at large $N$ and large 't Hoof coupling [3–7].

Recently, further progress was made on gauge theories that preserve a maximal amount of supersymmetry in dimensions other than four [8, 9]. In [8] the supergravity duals, including their ten-dimensional uplifts, were constructed for theories sourced by a stack of Euclidean spherical $p$-branes for $1 \leq p \leq 6$. The spherical branes have Euclidean $SU(N)$ gauge theories living on them, so presumably the supergravity solutions are the gravity duals for these gauge theories on $\mathbb{S}^{p+1}$. In [9] the free energies and expectation values of BPS Wilson loops were computed using localization and were shown to match at strong coupling with the corresponding quantities derived from the supergravity solutions in [8], up to possible identifiable counter terms.

The most intriguing result was the $p = 6$ case, where the gauge theory is on $\mathbb{S}^7$ with radius $\mathcal{R}$. Seven turns out to be the largest dimension where one can preserve the supersymmetry on the sphere [10, 11]. Here the "strongly" coupled gauge theory corresponded to taking the inverse effective 't Hooft coupling $\lambda_{\text{eff}}^{-1} \equiv \frac{\mathcal{R}^3}{g_{\text{YM}}^2 N}$ from $+\infty$, which corresponds to the true weak coupling limit, through the normal strong coupling point at $\lambda_{\text{eff}}^{-1} = 0$ to the negative side. The match with supergravity then occurs as $\lambda_{\text{eff}}^{-1} \to -\infty$, if one also analytically continues the supergravity solution such that the dictionary flips the sign in the relation of the string coupling to the Yang-Mills coupling.

A negative coupling is usually ill-defined in a gauge theory, but a similar phenomenon occurs in five dimensions which has a well understood physical interpretation [12]. Consider an $\mathcal{N} = 1$ $SU(N)$ gauge theory in five dimensions with an adjoint hypermultiplet with mass $M$. In this situation the gauge coupling is renormalized to

$$\frac{4\pi^2}{g_{\text{YM}}^2} = \frac{4\pi^2}{g_0^2} - M N, \tag{1.1}$$

where $g_0$ is the coupling that appears in the bare Lagrangian. If we take $M \to \infty$ then the

hypermultiplet decouples and we have a pure $\mathcal{N} = 1$ theory at energy scales much lower than $M$. As one varies $M$, one can tune $g_0$ to keep $g_{\text{YM}}$ fixed. Even with a positive $g_0$, we can tune $g_{\text{YM}}$ to be negative. To probe the phases of this theory it is more appropriate to consider the effective coupling $g_{\text{eff}}^2 = g_{\text{YM}}^2 E$, where $E$ is the energy scale. The $SU(N)$ gauge theory flows to one of three different phases: the normal weakly coupled phase where $\frac{1}{g_{\text{eff}}^2} \to \infty$; the UV fixed-point phase at $\frac{1}{g_{\text{eff}}^2} = 0$,[1] where effective field theory breaks down and there is a nontrivial superconformal field theory (SCFT) [12, 14]; and a different field theory phase at $\frac{1}{g_{\text{eff}}^2} \to -\infty$, which could include five-dimensional Chern-Simons theories [15, 16]. For example, for an $SU(2N)$ gauge group at weak negative coupling the theory is equivalent to an $SU(N)_N \times SU(N)_{-N} \times SU(2)$ theory, where the $SU(N)$ theories are pure Chern-Simons at levels $\pm N$, and the $SU(2)$ is an ordinary weakly coupled Yang-Mills theory. One thing that distinguishes the different weak coupling phases are the massless particles. In the positive coupling phase the theory has massless $W$-bosons and instanton particles with mass $m_I = \frac{4\pi^2}{g_{\text{YM}}^2}$. In the negative coupling phase the instantons are massless and exchange their role with some of the $W$-bosons.

Like this five-dimensional example, the gauge coupling of seven-dimensional SYM runs linearly with the scale. Hence, the $\lambda_{\text{eff}}$ relevant for the localized partition function as well as for the gauge-gravity dictionary is related to the bare coupling $g_0$ by

$$\lambda_{\text{eff}}^{-1} \equiv \frac{\mathcal{R}^3}{g_{\text{YM}}^2 N} = \frac{\mathcal{R}^3}{g_0^2 N} - \frac{n_0}{2\pi^4}, \tag{1.2}$$

where $n_0 \gg 1$ is a cutoff of the spherical harmonic modes. Cutting off these modes corresponds to imposing a UV cutoff $\Lambda$, which we can relate to $n_0$ by $n_0 = \Lambda \mathcal{R}$. The cutoff $\Lambda$ could represent the string scale or an eleven-dimensional Planck scale. As in the five-dimensional example, the effective coupling can be negative for various ranges of the parameters, even assuming a positive bare coupling.

However, unlike five dimensions, the sign of the effective coupling can depend on $\mathcal{R}$ for fixed $g_0$ and $\Lambda$, since their contributions in (1.2) come with different powers of $\mathcal{R}$. This means that $\lambda_{\text{eff}}^{-1} = 0$ cannot be a UV fixed point since the position of the zero is $\mathcal{R}$ dependent. Of course, this is a good thing since seven-dimensional SCFTs do not exist [17]. We then essentially have only two phases, the normal weakly coupled SYM as $\lambda_{\text{eff}}^{-1} \to \infty$, and the more mysterious phase as $\lambda_{\text{eff}}^{-1} \to -\infty$. Any consistent UV completion of the gauge theory is expected to include gravity, and the main outcome of this paper suggests that the phase at $\lambda_{\text{eff}}^{-1} = -\infty$ is also gravitational. This is in line with the prediction by Peet and Polchinski that there are two distinct phases for SYM in seven dimensions, one of them being gravitational [18]. The relation in (1.2) shows that while the $\lambda_{\text{eff}}^{-1} < 0$ phase is at a higher energy scale than the normal weak coupling phase, we can still reach this phase at a scale well below the cutoff $\Lambda$. In fact this condition is required in our analysis.

We can analyze these different regimes by studying the localized partition functions of the gauge theories on spheres, where the different phases are smoothly connected. For instance, one can study the five-dimensional SCFTs by setting $\frac{1}{g_{\text{eff}}^2} = 0$ in the localized theory, as was done to great effect in [19]. Likewise, one can study the phase at weak negative coupling by looking for the emergence of massless particles which are distinct from those in the normal weak coupling phase. In the five-dimensional gauge theories these are the instantons.

Localization reduces the functional integrals to matrix integrals whose integrands decompose into a classical, a perturbative, and a non-perturbative contribution. For the five-

---

[1]The term "UV fixed-point" is somewhat of a misnomer. A more proper way to think of it is as a nontrivial IR fixed point that has the Yang-Mills Lagrangian as a relevant operator (c.f. [13]).

dimensional theory on a squashed $\mathbb{S}^5$, the non-perturbative contributions come from the world-lines of point-like instantons that extend along circles fibered over the fixed-points of a two-dimensional complex base. At weak positive coupling these contributions are suppressed by factors of $\exp\left(-\frac{4\pi^2 n_I}{g_{\text{YM}}^2}\frac{2\pi\mathcal{R}}{\omega_i}\right)$, where $\mathcal{R}$ is the radius of the $\mathbb{S}^5$, $n_I$ is the instanton number, and $\omega_i$ is one of the three squashing parameters. This is the world-line contribution for $n_I$ instanton particles with mass $m_I$ on a fixed circle. If $g_{\text{YM}}^2$ is large and negative, then the instanton contribution appears to be exponentially enhanced. However, to see the emergence of the massless instantons one needs to consider the contribution of the Nekrasov partition function, which is derived from the quantum mechanics of the ADHM variables. Here one finds that the negative instanton mass is canceled off by an exponentially suppressed piece from the Nekrasov partition function, leaving a contribution one would expect from a massless particle.

In this paper we show that a similar phenomenon happens for supersymmetric Yang-Mills on $\mathbb{S}^7$. If one examines the perturbative contribution to the partition function, one sees that it has qualitatively the same behavior as in our five-dimensional example. One can then ask if there is similar behavior when it comes to the instantons when crossing from positive to negative inverse coupling. Of course, in five dimensions the usual co-dimension four instantons are particles while in seven dimensions they are membranes with tension $T_I = \frac{4\pi^2}{g_{\text{YM}}^2}$, so the seven-dimensional case will not have instantons and $W$-bosons exchanging roles. However, one might still expect a cancelation of the negative tension of the membranes from the analog of a Nekrasov partition function once the threshold between positive and negative inverse coupling is crossed. We will show that this is precisely what happens.

In order to regulate divergences it is necessary to consider the squashed sphere and then take the round sphere limit. The instanton contributions come from six $\mathbb{S}^3$ subspaces of the $\mathbb{S}^7$. In the round sphere limit the $\mathbb{S}^7$ is familiarly written as an $\mathbb{S}^1$ fibration over a $\mathbb{CP}^3$ base. The six $\mathbb{S}^3$ subspaces are the $\mathbb{S}^1$ fiber over six $\mathbb{CP}^1$ subspaces of $\mathbb{CP}^3$. For each $\mathbb{S}^3$ the instantons are point-like on the co-dimension four space transverse to the $\mathbb{S}^3$. Hence, their contribution to the partition function will be as a three-dimensional field theory for the ADHM variables. The structure of the instanton contributions are similar to the four- and five-dimensional cases [20, 21] and contain a sum over colored partitions represented as sums over Young tableaux. The various terms in the sum each come with an overall factor of $\exp\left(-\frac{4\pi^2}{g_{\text{YM}}^2}\text{vol}(\mathbb{S}^3)n_I\right)$, where $n_I$ is the instanton number and $\text{vol}(\mathbb{S}^3)$ is the volume of the $\mathbb{S}^3$. Like the five-dimensional case, when the inverse coupling is large and negative this exponential enhancement is canceled by the instanton partition function and the resulting factor is that of a membrane with tension $T = \frac{2}{\pi R^3}\delta\sigma$, where $\delta\sigma$ is a dimensionless scalar field. Hence, at the origin of $\delta\phi$ the tension is zero.

In five dimensions the instantons couple to a $U(1)$ gauge field which is part of a vector multiplet. Hence, in the weak negative phase this $U(1)$ is enhanced to an $SU(2)$. In seven dimensions, if we follow our five-dimensional example, the light instanton membranes in the weak negative phase should couple to a three-form field. The only supermultiplet containing this field is the gravity multiplet, hence we expect this phase to consist of weakly coupled gravity.

A useful check on our formalism is the cancelation of divergences. The instanton contribution at each $\mathbb{S}^3$ is singular in the round sphere limit. This is an indication that at the limit the localization locus can move off the six three-spheres. Nevertheless, one should expect that the overall partition function remain finite in the limit. We will see that at the one instanton level the singularities are fourth order poles and indeed we find that the singularities cancel when summing up their contribution over all six three-spheres.

The rest of the paper is organized as follows. In section 2 we review previous localization results for round and squashed seven-spheres. In section 3 we consider the localized partition

function for pure $\mathcal{N} = 1$ gauge theory on the squashed five-sphere, in particularly examining the behavior of the theory when the effective squared coupling is negative. In section 4 we study instanton contributions on the squashed seven-sphere, constructing an explicit partition function that includes these contributions. Specializing to an $SU(2)$ gauge group we show that that negative tension of the instanton membranes is canceled by contributions from the instanton partition function and that the tension then has a simple dependence on a scalar field. In section 5 we conclude with some speculations about the nature of the $SU(2)$ gauge theory at weak negative coupling and its relation to seven-dimensional minimal supergravity. The two appendices contain several technical details regarding the instanton contributions.

## 2 Preliminaries

In this section we study the SYM on the round and the squashed $\mathbb{S}^7$. We review the construction and the localization of the theory. We extend these results to compute the perturbative partition function for small and negative 't Hooft coupling.

We realize $\mathbb{S}^{2r-1}$ by an embedding in $\mathbb{R}^{2r} = \mathbb{C}^r$ given by

$$\sum_{i=1}^{r} |z|_i^2 = 1, \tag{2.1}$$

where $z_i$ are complex. The metric on the squashed sphere can be expressed as

$$ds_{\mathbb{S}^{2r-1}}^2 = \mathcal{R}^2 \sum_{i=1}^{r} \left( d\rho_i^2 + \rho_i^2 d\phi_i^2 \right) + \mathcal{R}^2 \frac{1}{1 - \sum_{i=1}^{r} a_i^2 \rho_i^2} \left( \sum_{i=1}^{r} a_i \rho_i^2 d\phi_i \right)^2, \tag{2.2}$$

where $z_i = \rho_i e^{i\phi_i}$ and $\omega_i = 1 + a_i$ are the squashing parameters. We further require that $\sum_i a_i = 0$. $\mathbb{S}^{2r-1}$ can be seen as a fibration over $\mathbb{CP}^{r-1}$ and this condition ensures that the squashing acts only on the base $\mathbb{CP}^{r-1}$ [22]. For a physical squashed sphere $\omega_i \in \mathbb{R}_+$, but it will be necessary to give them a small imaginary piece in order to have a well behaved partition function.

### 2.1 The round sphere

For the round sphere, where all $\omega_i = 1$, we can place the SYM on $\mathbb{S}^7$ while preserving 16 supersymmetries [11]. In this case we consider the Killing vector, $v = \sum_{i=1}^{4} \partial_{\phi_i}$ which is constructed from a Killing spinor $\xi$, $v = \xi \Gamma^\mu \xi \partial_\mu$. The round sphere has a contact structure with contact form $\kappa$, where $v$ acts as the corresponding Reeb vector satisfying $\iota_v \kappa = 1$.

The partition function can be obtained by localizing w.r.t to the supersymmetries generated by $\xi$. The localization locus is given by

$$\begin{aligned}
v^\mu F_{\mu\nu} &= 0, \\
v^\sigma H_{\sigma\mu\nu\lambda} &= 0, \\
D_\mu \phi_0 &= 0, \qquad K^m = -\frac{4}{r} \phi_0 (v_m \Lambda \epsilon), \\
\widehat{F}_{\mu\nu}^+ &= D_\sigma \Phi_{\mu\nu}{}^\sigma, \\
f &= -\tfrac{1}{12} [\Phi_{\mu\nu\lambda}, \Phi^{\mu\nu}{}_\sigma] d\kappa^{\lambda\sigma}.
\end{aligned} \tag{2.3}$$

Here $K^m$, $m = 1\ldots7$ are auxilary fields for the seven dimensional vector multiplet, $\phi_0$ is one of the three scalar fields that make up the vector multiplet, $\Phi_{\mu\nu\lambda}$ are three forms made from

the other two scalar fields,

$$\Phi_{\mu\nu\lambda} = \frac{1}{2}\phi_A\left(\xi\Gamma_{\mu\nu\lambda}\Gamma^{A0}\xi\right),\tag{2.4}$$

and $H$ is the field strength for $\Phi$,

$$H_{\sigma\mu\nu\lambda} \equiv D_\sigma\Phi_{\mu\nu\lambda} - D_\mu\Phi_{\sigma\nu\lambda} - D_\nu\Phi_{\mu\sigma\lambda} - D_\lambda\Phi_{\mu\nu\sigma}.\tag{2.5}$$

The field strength has been decomposed into a vertical and horizontal part, $F = F_V + F_H$, with $\kappa \wedge \iota_\nu F = F_V$ and

$$F_H = \widehat{F}^+ + \widehat{F}^- - \frac{1}{12}f\,\mathrm{d}\kappa.\tag{2.6}$$

The $\widehat{F}^\pm$ components are defined by

$$\widehat{F}^\pm = \pm\iota_\nu * \left(-\frac{1}{2}\widehat{F}^\pm \wedge \mathrm{d}\kappa\right).\tag{2.7}$$

Because of the contact structure, the horizontal space has an almost complex structure, such that $\Phi$ decomposes into $(3,0)$ and $(0,3)$ forms $\Phi^\pm$, while $\widehat{F}^+$ decomposes into $(2,0)$ and $(0,2)$ forms, and $\widehat{F}^-$ and $\breve{F} \equiv -\frac{1}{2}f\,\mathrm{d}\kappa$ are $(1,1)$ forms. In terms of the contact structure we can rewrite (2.3) as [23]

$$\begin{aligned}
\iota_\nu F &= 0,\\
\iota_\nu \mathrm{d}_A\Phi &= 0,\\
\mathrm{d}_A\phi_0 &= 0,\\
\widehat{F}^+ &= *\mathrm{d}_A*\Phi,\\
\breve{F}\wedge\mathrm{d}\kappa\wedge\mathrm{d}\kappa &= 4[\Phi^-,\Phi^+],
\end{aligned}\tag{2.8}$$

where $\Phi^+$ is the $(3,0)$ form and $\Phi^-$ is the $(0,3)$ form. The bottom three equations are those of the six-dimensional Hermitian Higgs-Yang-Mills equations discussed in [24].

The perturbative contribution to the partition function has $F = \Phi = 0$ and $\phi_0$ constant. It is given by [11]

$$\mathcal{Z}_{\mathrm{pert}} = \int \prod_{i=1}^{N}\mathrm{d}\sigma_i\,\delta\left(\sum_i\sigma_i\right)e^{-\frac{4\pi^4\mathcal{R}^3}{g_0^2}\sum_i\sigma_i^2}\prod_{i<j}^{N}\prod_{n=-\infty}^{\infty}\left\{(n^2+\sigma_{ij}^2)^{n^2+1}\right\},\tag{2.9}$$

where $\sigma_i$ are the eigenvalues of $\mathcal{R}\phi_0$, $\sigma_{ij} \equiv \sigma_i - \sigma_j$ and $g_0^2$ is the bare coupling that appears in the Lagrangian. This partition function is divergent and needs to be regularized [9]. This can be accomplished by multiplying and dividing by $e^{-\sigma_{ij}^2}$ within the curly brackets and then instituting a cutoff, $n_0 = \Lambda\mathcal{R}$ in the mode numbers $n$. The cutoff is justified if the dominant contributions to the partition function come from the integration regions where $|\sigma_i| \ll n_0$. Using that

$$\sum_{i<j}\sigma_{ij}^2 = N\sum_i\sigma_i^2 - \left(\sum_i\sigma_i\right)^2 = N\sum_i\sigma_i^2,\tag{2.10}$$

we can rewrite the perturbative partition function as

$$\mathcal{Z}_{\mathrm{pert}} = \int \prod_{i=1}^{N}\mathrm{d}\sigma_i\,\delta\left(\sum_i\sigma_i\right)e^{-\frac{4\pi^4\mathcal{R}^3}{g_{\mathrm{YM}}^2}\sum_i\sigma_i^2}\prod_{i<j}^{N}\prod_{n=-\infty}^{\infty}\left\{(n^2+\sigma_{ij}^2)^{n^2+1}e^{-\sigma_{ij}^2}\right\},\tag{2.11}$$

where $g_{\mathrm{YM}}^2$ is the renormalized coupling satisfying

$$\frac{4\pi^4 \mathcal{R}^3}{g_{\mathrm{YM}}^2} = \frac{4\pi^4 \mathcal{R}^3}{g_0^2} - 2N\Lambda\mathcal{R}\,. \tag{2.12}$$

Note that while the bare coupling is positive definite, the renormalized coupling could be negative.

We can solve the partition function in (2.11) by saddle point [9]. Extremizing the partition function leads to the equations

$$\frac{8\pi^4}{\lambda}\sigma_i = \frac{2\pi}{N}\sum_{j\neq i}(1-\sigma_{ij}^2)\coth\pi\sigma_{ij}\,, \tag{2.13}$$

where $\lambda$ is the dimensionless 't Hooft coupling $\lambda = g_{\mathrm{YM}}^2 N \mathcal{R}^{-3}$. These equations are very similar to those found for pure $\mathcal{N}=1$ super Yang-Mills on $S^5$ [25] so we can borrow many of the methods from there. We are most interested in having $\lambda$ be small and negative. To simplify the discussion we assume that $N$ is even. In this case the solution to (2.13) can be well approximated by

$$\begin{aligned}\sigma_i &= \sigma_0 + \delta\sigma_i\,, & 1 \leq i \leq N/2\,, \\ \sigma_{i+N/2} &= -\sigma_0 + \delta\tilde{\sigma}_i\,,\end{aligned} \tag{2.14}$$

where we choose

$$\sum_i^{N/2}\delta\sigma_i = \sum_i^{N/2}\delta\tilde{\sigma}_i = 0\,. \tag{2.15}$$

(2.13) then becomes

$$\frac{8\pi^4 N}{\lambda}(\sigma_0 + \delta\sigma_i) = \pi\sum_{j\neq i}^{N/2}\left(2 - 2(\delta\sigma_i - \delta\sigma_j)^2\right)\coth(\pi(\delta\sigma_i - \delta\sigma_j)) \tag{2.16}$$

$$+ \pi N - \pi N\left(4\sigma_0^2 + 4\sigma_0\delta\sigma_i + (\delta\sigma_i)^2\right) - 2\pi\sum_{j=1}^{N/2}(\delta\tilde{\sigma}_j)^2 + \mathrm{O}(e^{-2\pi\sigma_0})\,.$$

If we sum (2.16) over $i$, we then find

$$4\sigma_0^2 + \frac{8\pi^3}{\lambda}\sigma_0 - 1 + \overline{\delta\sigma^2} + \overline{\delta\tilde{\sigma}^2} = \mathrm{O}(e^{-2\pi\sigma_0})\,, \tag{2.17}$$

where the averaged squares are defined as

$$\overline{\delta\sigma^2} \equiv \frac{2}{N}\sum_{i=1}^{N/2}\delta\sigma_i^2\,, \qquad \overline{\delta\tilde{\sigma}^2} \equiv \frac{2}{N}\sum_{i=1}^{N/2}\delta\tilde{\sigma}_i^2\,. \tag{2.18}$$

Dropping the exponentially suppressed term, we find for $\lambda \to 0_-$

$$\sigma_0 \approx -\frac{2\pi^3}{\lambda} - \frac{\lambda}{8\pi^3}\delta_0\,, \tag{2.19}$$

where

$$\delta_0 = (1 - \overline{\delta\sigma^2} - \overline{\delta\tilde{\sigma}^2})\,. \tag{2.20}$$

Substituting back into (2.16) we end up with

$$\pi N\left(\delta\sigma_i^2 - \frac{\lambda}{2\pi^3}\delta_0\delta\sigma_i - \overline{\delta\sigma^2}\right) = \pi\sum_{j\neq i}^{N/2}\left(2 - 2(\delta\sigma_i - \delta\sigma_j)^2\right)\coth(\pi(\delta\sigma_i - \delta\sigma_j)), \quad (2.21)$$

One finds an analogous equation for $\delta\tilde{\sigma}_i$. Note that the validity of the cutoff requires that $\sigma_0 \ll n_0$. Hence, the bare coupling in (2.12) needs to be tuned in order that $1 \ll -\frac{2\pi^4}{\lambda} \ll n_0$.

As in the five-dimensional case, the term linear in $\delta\sigma_i$ can be dropped for small negative $\lambda$. Moreover, in the large $N$ limit $\delta_0$ is suppressed by a factor of $1/N$ [25]. The remaining terms on the left hand side of (2.21) can be generated by the free energy

$$F = \pi N\sum_{i=1}^{N/2}\left(\frac{1}{3}(\delta\sigma_i)^3 + \chi(\delta\sigma_i)\right), \quad (2.22)$$

where $\chi = -\overline{\delta\sigma^2}$. We can interpret the $\delta\sigma_i$ as the eigenvalues for an adjoint scalar in the vector multiplet of an SU($N/2$) gauge theory. The first term in $F$ originates from the term in the effective action

$$i\frac{N}{48\pi^3}\int \text{Tr}(\sigma^3)\kappa \wedge d\kappa \wedge d\kappa \wedge d\kappa, \quad (2.23)$$

where we used that the volume form is given by $\text{Vol} = -\frac{1}{48}\kappa \wedge (d\kappa)^3$. (2.23) is part of the supersymmetric completion [23] of

$$\frac{N}{2}\int c_3(A) \wedge \kappa, \quad (2.24)$$

where $c_3(A)$ is the third Chern character

$$c_3(A) = \frac{1}{24\pi^3}\text{Tr}(F \wedge F \wedge F). \quad (2.25)$$

(2.24) is topological and equals $Nk\pi$ where $k$ is an integer. Since we have assumed that $N$ is even this is a multiple of $2\pi$ and will not contribute to the partition function.

## 2.2 The squashed sphere

Under a general squashing, the sphere is a Sasaki-Einstein manifold which preserves two supersymmetries [23, 26]. In this case the Reeb vector is

$$v = \sum_{i=1}^{4}\omega_i\partial_{\phi_i}, \quad (2.26)$$

which for general $\omega_i$ does not generate closed orbits except in isolated cases. Localizing with respect to the Killing spinor that generates (2.26) one finds the same form for the localization locus as in (2.8) and a perturbative partition function given by [11, 23]

$$\mathcal{Z}_{\text{pert}} = \int\prod_{i=1}^{N}d\sigma_i e^{-\frac{4\pi^4\mathcal{R}^3\varrho}{g_{\text{YM}}^2}\sum_i\sigma_i^2}\prod_{i<j}^{N}S_4(i\sigma_{ij};\omega_1,\omega_2,\omega_3,\omega_4)$$
$$\times S_4(-i\sigma_{ij};\omega_1,\omega_2,\omega_3,\omega_4), \quad (2.27)$$

where $S_4(x; \omega_1, \omega_2, \omega_3, \omega_4)$ is the quadruple sine, which in unregularized form is given by

$$S_4(z; \omega_1, \omega_2, \omega_3, \omega_4) = \frac{\displaystyle\prod_{i,j,k,l=0}^{\infty} (i\omega_1 + j\omega_2 + k\omega_3 + l\omega_4 + z)}{\displaystyle\prod_{i,j,k,l=1}^{\infty} (i\omega_1 + j\omega_2 + k\omega_3 + l\omega_4 - z)}, \qquad (2.28)$$

and $\varrho = (\omega_1 \omega_2 \omega_3 \omega_4)^{-1}$ is the volume ratio of the squashed to the round sphere.

We are again interested in small negative $g_{\text{YM}}^2$, where the eigenvalues split into two groups separated far from each other. In this case, it is convenient to write the quadruple sine in product form [27],

$$
\begin{aligned}
S_4(z; \vec{\omega}) &= \exp\left(\frac{\pi i}{24} B_{44}(z; \vec{\omega})\right) \\
&\times \prod_{j,k,l \geq 0}^{\infty} \frac{\left(1 - e^{2\pi i\left(\frac{z}{\omega_4} + j\frac{\omega_1}{\omega_4} + k\frac{\omega_2}{\omega_4} + l\frac{\omega_3}{\omega_4}\right)}\right)\left(1 - e^{2\pi i\left(\frac{z}{\omega_2} + j\frac{\omega_1}{\omega_2} - (k+1)\frac{\omega_3}{\omega_2} - (l+1)\frac{\omega_4}{\omega_2}\right)}\right)}{\left(1 - e^{2\pi i\left(\frac{z}{\omega_3} + j\frac{\omega_1}{\omega_3} + k\frac{\omega_2}{\omega_3} - (l+1)\frac{\omega_4}{\omega_3}\right)}\right)\left(1 - e^{2\pi i\left(\frac{z}{\omega_1} - (j+1)\frac{\omega_2}{\omega_1} - (k+1)\frac{\omega_3}{\omega_1} - (l+1)\frac{\omega_4}{\omega_1}\right)}\right)} \\
&= \exp\left(-\frac{\pi i}{24} B_{44}(z; \vec{\omega})\right) \\
&\times \prod_{j,k,l \geq 0}^{\infty} \frac{\left(1 - e^{-2\pi i\left(\frac{z}{\omega_1} + j\frac{\omega_4}{\omega_1} + k\frac{\omega_3}{\omega_1} + l\frac{\omega_2}{\omega_1}\right)}\right)\left(1 - e^{-2\pi i\left(\frac{z}{\omega_3} + j\frac{\omega_4}{\omega_3} - (k+1)\frac{\omega_2}{\omega_3} - (l+1)\frac{\omega_1}{\omega_3}\right)}\right)}{\left(1 - e^{-2\pi i\left(\frac{z}{\omega_2} + j\frac{\omega_4}{\omega_2} + k\frac{\omega_3}{\omega_2} - (l+1)\frac{\omega_1}{\omega_2}\right)}\right)\left(1 - e^{-2\pi i\left(\frac{z}{\omega_4} - (j+1)\frac{\omega_3}{\omega_4} - (k+1)\frac{\omega_2}{\omega_4} - (l+1)\frac{\omega_1}{\omega_4}\right)}\right)},
\end{aligned}
$$
$$(2.29)$$

where $B_{44}(z; \vec{\omega})$ is a multiple Bernoulli polynomial whose odd terms are given by

$$
\begin{aligned}
\frac{1}{2}\left(B_{44}(z; \vec{\omega}) - B_{44}(-z; \vec{\omega})\right) &= -\frac{2z^3 + z(\omega_1\omega_2 + \text{perms})}{\omega_1\omega_2\omega_3\omega_4}(\omega_1 + \omega_2 + \omega_3 + \omega_4) \\
&= -4\frac{2z^3 + z(\omega_1\omega_2 + \text{perms})}{\omega_1\omega_2\omega_3\omega_4}. \qquad (2.30)
\end{aligned}
$$

The products as written in (2.29) are well-defined only if the $\omega_i$ are given small imaginary pieces such that $\text{Im}(\omega_i/\omega_j) > 0$ (and so $\text{Im}(\omega_j/\omega_i) < 0$) if $i < j$. For large positive or negative imaginary $z$, we then see that

$$
\begin{aligned}
&\log\left(S_4(z; \vec{\omega}) S_4(-z; \vec{\omega})\right) \qquad\qquad\qquad\qquad\qquad\qquad\qquad\qquad (2.31) \\
&\approx -\frac{\pi\varrho}{3}\left(2|\text{Im}(z)|^3 - |\text{Im}(z)|(\omega_1\omega_2 + \omega_1\omega_3 + \omega_1\omega_4 + \omega_2\omega_3 + \omega_2\omega_4 + \omega_3\omega_4)\right).
\end{aligned}
$$

From this and (2.27) we learn that for small negative $\lambda$, the saddle point equation for the partition function (2.17) is modified in the squashed case to

$$
\begin{aligned}
&4\sigma_0^2 + \frac{8\pi^3}{\lambda}\sigma_0 - \frac{(\omega_1\omega_2 + \omega_1\omega_3 + \omega_1\omega_4 + \omega_2\omega_3 + \omega_2\omega_4 + \omega_3\omega_4)}{6} + \overline{\delta\sigma^2} + \overline{\delta\tilde{\sigma}^2} \\
&= O(e^{-2\sigma_0}). \qquad\qquad\qquad\qquad\qquad\qquad\qquad\qquad\qquad\qquad\qquad\qquad (2.32)
\end{aligned}
$$

The perturbative partition function is well-defined for small coupling. For positive couplings, the non-perturbative effects are exponentially suppressed. For negative $\lambda$, instantons can no longer be ignored and one needs their contribution to find the total partition function. We will return to this problem in section 4.

## 3 A five-dimensional detour

Before confronting the problem of negative 't Hooft coupling $\lambda$, let us first consider the similar problem for 5d $\mathcal{N} = 1$ SYM with an adjoint hypermultiplet. In this case we have an understanding of what happens at negative couplings due to [12]. Our aim here is to explore the negative coupling regime using localization and learn lessons which will then be applied to the seven dimensional case.

### 3.1 The localized partition function

On $\mathbb{S}^5$ localizing with respect to the Killing spinor that generates the Reeb vector one finds for the locus [28]

$$
\begin{aligned}
\iota_v(*F) &= -F, \\
d_A\phi &= 0,
\end{aligned}
\tag{3.1}
$$

where $\phi$ is the real adjoint scalar that is part of the vector multiplet. Notice that the first equation also implies that $\iota_v F = 0$. It then follows that $\mathcal{L}_v A = d(\iota_v A) + [A, \iota_v A]$, i.e. the Lie derivative of the gauge field $A$ along $\mathbf{v}$ is a gauge transformation. As in the case for seven dimensions, the orbit generated by $\mathbf{v}$ does not close for generic toric data $(|z_1|, |z_2|, |z_3|)$ and squashing parameters $\omega_i$. In this case the orbit is dense on the three-torus over $(|z_1|, |z_2|, |z_3|)$. This suggests that for these orbits, in order to avoid singular configurations $\iota_v F = 0$ implies $F = 0$ [29]. Hence, the only nontrivial contributions can occur at the fixed points $(|z_1|, |z_2|, |z_3|) = (1, 0, 0), (0, 1, 0)$, or $(0, 0, 1)$ where the orbits close and there can be point-like instantons along the orbit. The space transverse to each closed orbit can be replaced with a $\mathbb{C}^2$, and in circling the orbit the transverse space is twisted. For example, at $(0, 0, 1)$ we have that $(z_1, z_2) \to (z_1 e^{2\pi i \frac{\omega_1}{\omega_3}}, z_2 e^{2\pi i \frac{\omega_2}{\omega_3}})$. Hence, the nonperturbative contribution from each fixed point to the partition function is the Nekrasov partition function [20, 21], $Z_{\text{inst}}(i\sigma, i\mu, \beta_i, \epsilon_{1,i}, \epsilon_{2,i})$, where $\beta_i = \frac{2\pi}{\omega_i}$, the equivariant parameters are given by the other two squashing parameters and $\mu = M\mathcal{R}$ where $M$ is the mass of the hypermultiplet.

This then fits with the factorization hypothesis in [30], where the authors exploited the resemblance of the perturbative partition function to partition functions in topological string theory to conjecture a full nonperturbative partition function on the squashed sphere. For the case with the adjoint hypermultiplet this is given by

$$
\begin{aligned}
\mathcal{Z} = \int \prod_{i=1}^{N} d\sigma_i\, &e^{-\frac{4\pi^3 \mathcal{R} \varrho_5}{g_0^2} \sum_i \sigma_i^2} \\
&\times \prod_{i<j}^{N} \frac{S_3(i\sigma_{ij}; \omega_1, \omega_2, \omega_3) S_3(-i\sigma_{ij}; \omega_1, \omega_2, \omega_3)}{S_3(i\sigma_{ij} + \frac{\Delta}{2} + i\mu; \omega_1, \omega_2, \omega_3) S_3(-i\sigma_{ij} + \frac{\Delta}{2} + i\mu; \omega_1, \omega_2, \omega_3)} \\
&\times Z_{\text{inst}}\left(i\sigma, i\mu, \frac{2\pi}{\omega_1}, \omega_2, \omega_3\right) Z_{\text{inst}}\left(i\sigma, i\mu, \frac{2\pi}{\omega_2}, \omega_3, \omega_1\right) \\
&\times Z_{\text{inst}}\left(i\sigma, i\mu, \frac{2\pi}{\omega_3}, \omega_1, \omega_2\right),
\end{aligned}
\tag{3.2}
$$

where $\Delta = \omega_1 + \omega_2 + \omega_3 = 3$, $\varrho_5 = (\omega_1 \omega_2 \omega_3)^{-1}$ and $S_3(x; \vec{\omega})$ is the triple sine, which we can

write as

$$
\begin{aligned}
S_3(z; \vec{\omega}) &= \exp\left(-\frac{\pi i}{6} B_{33}(z; \vec{\omega})\right) \\
&\quad \times \prod_{j,k \geq 0}^{\infty} \frac{\left(1 - e^{2\pi i\left(\frac{z}{\omega_3} + j\frac{\omega_1}{\omega_3} + k\frac{\omega_2}{\omega_3}\right)}\right)\left(1 - e^{2\pi i\left(\frac{z}{\omega_1} - (j+1)\frac{\omega_2}{\omega_1} - (k+1)\frac{\omega_3}{\omega_1}\right)}\right)}{\left(1 - e^{2\pi i\left(\frac{z}{\omega_2} + j\frac{\omega_1}{\omega_2} - (k+1)\frac{\omega_3}{\omega_2}\right)}\right)} \\
&= \exp\left(+\frac{\pi i}{6} B_{33}(z; \vec{\omega})\right) \\
&\quad \times \prod_{j,k \geq 0}^{\infty} \frac{\left(1 - e^{-2\pi i\left(\frac{z}{\omega_1} + j\frac{\omega_3}{\omega_1} + k\frac{\omega_2}{\omega_1}\right)}\right)\left(1 - e^{-2\pi i\left(\frac{z}{\omega_3} - (j+1)\frac{\omega_2}{\omega_3} - (k+1)\frac{\omega_1}{\omega_3}\right)}\right)}{\left(1 - e^{-2\pi i\left(\frac{z}{\omega_2} + j\frac{\omega_3}{\omega_2} - (k+1)\frac{\omega_1}{\omega_2}\right)}\right)} , \quad (3.3)
\end{aligned}
$$

with $B_{33}(z; \vec{\omega})$ given by

$$
\begin{aligned}
B_{33}(z; \vec{\omega}) &= \frac{z^3}{\omega_1 \omega_2 \omega_3} - \frac{3(\omega_1 + \omega_2 + \omega_3)}{2\omega_1 \omega_2 \omega_3} z^2 + \frac{\omega_1^3 + \omega_2^3 + \omega_3^3 + 3(\omega_1 \omega_2 + \omega_2 \omega_3 + \omega_3 \omega_1)}{2\omega_1 \omega_2 \omega_3} z \\
&\quad - \frac{(\omega_1 + \omega_2 + \omega_3)(\omega_1 \omega_2 + \omega_2 \omega_3 + \omega_3 \omega_1)}{4\omega_1 \omega_2 \omega_3} . \quad (3.4)
\end{aligned}
$$

The explicit instanton partition functions are given by [20, 21]

$$
Z_{\text{inst}}\left(i\sigma, i\mu, \frac{2\pi}{\omega_1}, \omega_2, \omega_3\right) = \sum_{\vec{Y}} e^{-\frac{4\pi^2 |\vec{Y}|}{g_0^2} \frac{2\pi \mathcal{R}}{\omega_1}} \prod_{i,j=1}^{N} \prod_{s \in Y_i} \Bigg\{
$$

$$
\frac{S_1\left(i\sigma_{ji} + \frac{\Delta}{2} + i\mu - (v_i(s)+1)\omega_2 + h_j(s)\omega_3; \omega_1\right) S_1\left(i\sigma_{ij} + \frac{\Delta}{2} + i\mu - (h_j(s)+1)\omega_3 + v_i(s)\omega_2; \omega_1\right)}{S_1\left(i\sigma_{ji} - (v_i(s)+1)\omega_2 + h_j(s)\omega_3; \omega_1\right) S_1\left(i\sigma_{ij} - (h_j(s)+1)\omega_3 + v_i(s)\omega_2; \omega_1\right)} \Bigg\},
$$

$$(3.5)$$

where $S_1(x, \omega)$ is the "single" sine function, defined as

$$
S_1(x, \omega) = \prod_{n=0}^{\infty} (n\omega + x) \prod_{n=1}^{\infty} (n\omega - x) = 2\sin\left(\frac{\pi}{\omega} x\right). \quad (3.6)
$$

The sum in (3.5) is over the colored partitions, with $\vec{Y}$ representing the $N$-tuple $\{Y_1, Y_2, \ldots Y_N\}$. Each $Y_i$ refers to a Young diagram, with $|Y_i|$ the number of boxes for that diagram, and $|\vec{Y}| = \sum_i |Y_i|$ which is the instanton number. The product over $s$ refers to each box in the particular diagram while $h_j(s)$ measures the horizontal distance to the edge for box $s$ in diagram $Y_j$, and $v_i(s)$ measures the vertical distance to the edge for box $s$ in diagram $Y_i$. Since $s \in Y_i$ is not necessarily a box in $Y_j$, $h_j(s)$ can be negative.

Let us take the adjoint mass parameter $\mu$ to infinity such that the theory reduces to a pure $\mathcal{N} = 1$ $SU(N)$ gauge theory. Using the product formulae in (3.3), up to an overall constant one can replace the partition function with

$$
\begin{aligned}
\mathcal{Z}' &= \int \prod_{i=1}^{N} d\sigma_i e^{-\frac{4\pi^3 \mathcal{R} \varrho_5}{g_{\text{YM}}^2} \sum_i \sigma_i^2} \prod_{i<j}^{N} S_3(i\sigma_{ij}; \omega_1, \omega_2, \omega_3) S_3(-i\sigma_{ij}; \omega_1, \omega_2, \omega_3) \\
&\quad \times Z'_{\text{inst}}\left(i\sigma, \frac{2\pi}{\omega_1}, \omega_2, \omega_3\right) Z'_{\text{inst}}\left(i\sigma, \frac{2\pi}{\omega_2}, \omega_3, \omega_1\right) Z'_{\text{inst}}\left(i\sigma, \frac{2\pi}{\omega_3}, \omega_1, \omega_2\right),
\end{aligned}
$$

$$(3.7)$$

where $\frac{4\pi^2}{g_{\text{YM}}^2} = \frac{4\pi^2}{g_0^2} - NM$ and

$$
Z'_{\text{inst}}\left(i\sigma, \frac{2\pi}{\omega_1}, \omega_2, \omega_3\right) = \sum_{\vec{Y}} e^{-\frac{4\pi^2|\vec{Y}|}{g_{\text{YM}}^2}\frac{2\pi\mathcal{R}}{\omega_1}} \prod_{i,j=1}^{N} \prod_{s\in Y_i} \left\{ \right.
$$

$$
\left. \frac{1}{S_1\big(i\sigma_{ji}-(v_i(s)+1)\,\omega_2+h_j(s)\omega_3;\omega_1\big)S_1\big(i\sigma_{ij}-\big(h_j(s)+1\big)\omega_3+v_i(s)\omega_2;\omega_1\big)} \right\}.
$$

(3.8)

### 3.2 The theory at negative coupling and instantons

As in the seven-dimensional case, we can have $g_{\text{YM}}^2 < 0$ in $5d$. But here we have an understanding of how to interpret the theory at negative coupling. For the sake of simplicity let us consider the case of $SU(2)$. At generic positive coupling there is a global $U(1)$ symmetry coming from the instanton current $\star(F \wedge F)$. On the Coulomb branch the mass of the $W$-bosons is $m_w = \phi$, where $\phi$ is the expectation value of the real adjoint scalar in the vector multiplet. The instantons have charges $\pm 1$ under the unbroken $U(1)$ gauge symmetry and their mass is $m_I = \phi + \frac{4\pi^2}{g_{\text{YM}}^2}$. At the origin of the Coulomb branch both the $W$-bosons and instantons become massless at infinite coupling and the global $U(1)$ symmetry is enhanced to $SU(2)$ [12,14,31]. Under a Weyl reflection for the global $SU(2)$ the parameters transform as $\frac{4\pi^2}{g_{\text{YM}}^2} \to -\frac{4\pi^2}{g_{\text{YM}}^2}$ and $\phi \to \phi - \frac{4\pi^2}{g_{\text{YM}}^2}$ and the roles of the $W$-bosons and instantons are reversed.

One can describe the above using $(p,q)$ webs of five branes [15,16], as shown in figure 1. Here the web has 4 fixed external branes with $(p,q)$ charges $(\pm 1, 1)$, along with two parallel NS5 branes and two parallel D5 branes orthogonal to the NS5 branes. The separation between these two sets of branes can change. Figure 1 (a) shows the positive coupling case. The $W$-bosons correspond to strings stretched between the two D5 branes. The separation of the branes $\phi$ leads to their mass. The instantons correspond to D1 branes stretched between the two NS5 branes whose separation is $\phi + \frac{4\pi^2}{g_{\text{YM}}^2}$. Figure 1 (b) shows the negative coupling case. Here the separation between the NS5 branes is $\phi$ and the separation between the D5 branes is $\phi - \frac{4\pi^2}{g_{\text{YM}}^2}$, so that the roles of the two particles have interchanged.

Because of the $SU(2)$ global symmetry at the superconformal fixed point, or by the $SL(2,Z)$ duality of the type IIB string theory that the $(p,q)$ branes live in, we see that the $SU(2)$ gauge theory with coupling $g_{\text{YM}}^2$ is equivalent to the one with coupling $-g_{\text{YM}}^2$. One should be able to see this in the partition function in (3.7). This is not obvious from the form of the instanton partition functions, but it is guaranteed to work from the conjectured structure of the partition function in [30] and the relations shown in [32] between the instanton and topological string partition functions.

Let us sketch how this works when the coupling is small but negative. At the saddle point the eigenvalue $\sigma_1 = -\sigma_2$ is large, hence we can make the approximation

$$
e^{-\frac{4\pi^3\mathcal{R}\varrho_5}{g_{\text{YM}}^2}\frac{\sigma_{12}^2}{2}} S_3(i\sigma_{12};\vec{\omega})S_3(-i\sigma_{12};\vec{\omega})
$$

$$
\approx \exp\left(-\frac{4\pi^3\mathcal{R}\varrho_5}{g_{\text{YM}}^2}\frac{\sigma_{12}^2}{2} - \frac{\pi\varrho_5}{3}\left(\sigma_{12}^3 - \frac{\omega_1^3+\omega_2^3+\omega_3^3+3(\omega_1\omega_2+\omega_2\omega_3+\omega_3\omega_1)}{2}\sigma_{12}\right)\right)
$$

$$
= C\exp\left(\frac{4\pi^3\mathcal{R}\varrho_5}{g_{\text{YM}}^2}\frac{\delta\sigma^2}{2} - \frac{\pi\varrho_5}{3}\left(\delta\sigma^3 - \frac{\omega_1^3+\omega_2^3+\omega_3^3+3(\omega_1\omega_2+\omega_2\omega_3+\omega_3\omega_1)}{2}\delta\sigma\right)\right),
$$

(3.9)

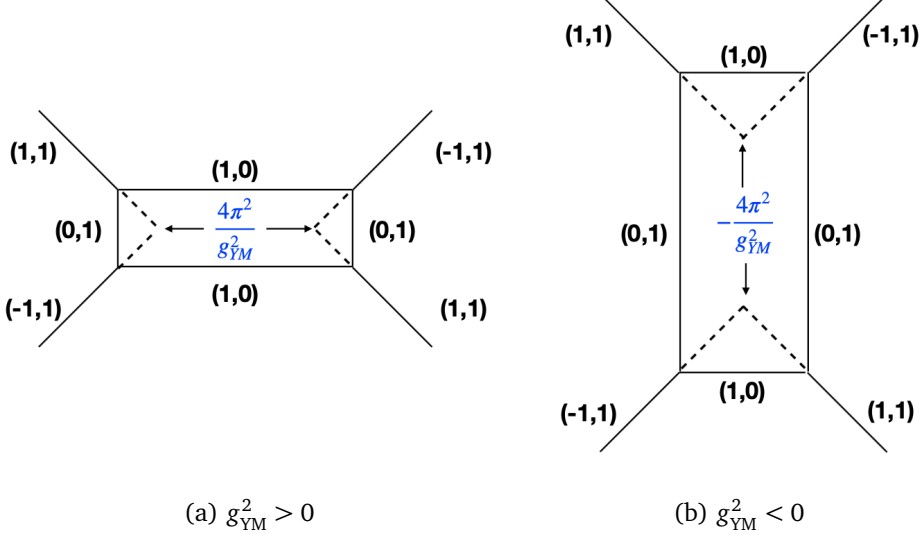

(a) $g_{\text{YM}}^2 > 0$        (b) $g_{\text{YM}}^2 < 0$

Figure 1: $(p,q)$ web for $\mathcal{N}=1$ SU(2) gauge theory at positive and negative coupling. D5 branes are $(1,0)$ branes and NS5 branes are $(0,1)$. The coupling is determined by the positions of the fixed $(\pm 1,1)$ external branes.

where $\delta\sigma = \sigma_{12} + \frac{4\pi^2 \mathcal{R}}{g_{\text{YM}}^2}$. Hence, the last line of (3.9) has the same form as the line above, except it has the opposite coupling term.

Let us now consider the instanton contribution. Note that the argument of the exponential in (3.8) is the negative of the world-line action of $|\vec{Y}|$ instanton particles with mass $\frac{4\pi^2}{g_{\text{YM}}^2}$ along the Reeb orbit. However, the mass is missing the contribution of the Coulomb branch which lurks in the rest of the expression in (3.8). To flesh this out, if we examine this expression we see that there are essentially two types of terms, depending on whether or not $i = j$. For a given $\vec{Y} = \{Y_1, Y_2\}$ the contribution from all terms where $i \neq j$ is

$$\prod_{s_1 \in Y_1} \frac{1}{4} \csc\left[\frac{\pi}{\omega_1}\left((v_1(s_1)+1)\,\omega_2 - h_2(s_1)\omega_3 + i\sigma_{12}\right)\right]$$

$$\times \csc\left[\frac{\pi}{\omega_1}\left((h_2(s_1)+1)\,\omega_3 - v_1(s_1)\omega_2 - i\sigma_{12}\right)\right]$$

$$\times \prod_{s_2 \in Y_2} \frac{1}{4} \csc\left[\frac{\pi}{\omega_1}\left((v_2(s_2)+1)\,\omega_2 - h_1(s_2)\omega_3 - i\sigma_{12}\right)\right]$$

$$\times \csc\left[\frac{\pi}{\omega_1}\left((h_1(s_2)+1)\,\omega_3 - v_2(s_2)\omega_1 + i\sigma_{12}\right)\right]$$

$$\approx \prod_{s_1 \in Y_1} \exp\left[\frac{\pi}{\omega_1}\left(i\,(2v_1(s_1)+1)\,\omega_2 - i\,(2h_2(s_1)+1)\omega_3 - 2\sigma_{12}\right)\right]$$

$$\times \prod_{s_2 \in Y_2} \exp\left[\frac{\pi}{\omega_1}\left(i\,(2h_1(s_2)+1)\,\omega_3 - i\,(2v_2(s_2)+1)\omega_2 - 2\sigma_{12}\right)\right]$$

$$= e^{\frac{8\pi^3 r |\vec{Y}|}{g_{\text{YM}}^2 \omega_1}} e^{-\frac{2\pi |\vec{Y}| \delta\sigma}{\omega_1}} \prod_{s_1 \in Y_1} \exp\left[\frac{\pi}{\omega_1} i\left((2v_1(s_1)+1)\,\omega_2 + (2h_1(s_1)+1)\omega_3\right)\right]$$

$$\times \prod_{s_2 \in Y_2} \exp\left[-\frac{\pi}{\omega_1} i\left((2h_2(s_2)+1)\,\omega_3 + (2v_2(s_2)+1)\omega_2\right)\right], \quad (3.10)$$

where we have again expanded around the saddle point $\sigma_{12} = -\frac{4\pi^2 \mathcal{R}}{g_{YM}^2}$ in the small negative

$g_{YM}^2$ limit. The last step involves the identity shown in appendix A. The contribution from the terms where $i = j$ is

$$\prod_{s_1 \in Y_1} \frac{1}{4} \csc\left[\frac{\pi}{\omega_1}\left((v_1(s_1)+1)\,\omega_2 - h_1(s_1)\omega_3\right)\right] \csc\left[\frac{\pi}{\omega_1}\left((h_1(s_1)+1)\,\omega_3 - v_1(s_1)\omega_2\right)\right]$$
$$\times \prod_{s_2 \in Y_2} \frac{1}{4} \csc\left[\frac{\pi}{\omega_1}\left((v_2(s_2)+1)\,\omega_2 - h_2(s_2)\omega_3\right)\right] \csc\left[\frac{\pi}{\omega_1}\left((h_2(s_2)+1)\,\omega_3 - v_2(s_2)\omega_2\right)\right]. \tag{3.11}$$

From (3.10) and (3.11) we see that we can write the instanton contribution in (3.8) as the factorized product

$$Z'_{\text{inst}}\left(i\sigma, \frac{2\pi}{\omega_1}, \omega_2, \omega_3\right) \approx \mathcal{Z}(\delta\sigma|\omega_1; \omega_2, \omega_3)\mathcal{Z}(\delta\sigma|\omega_1; -\omega_2, -\omega_3), \tag{3.12}$$

where

$$\mathcal{Z}(\delta\sigma|\omega_1; \omega_2, \omega_3)$$
$$= \sum_Y e^{-\frac{2\pi|Y|\delta\sigma}{\omega_1}} \prod_{s \in Y} \frac{\frac{1}{4}\exp\left[\frac{\pi}{\omega_1}i\left((2v(s)+1)\,\omega_2 + (2h(s)+1)\omega_3\right)\right]}{\sin\left[\frac{\pi}{\omega_1}\left((v(s)+1)\,\omega_2 - h(s)\omega_3\right)\right]\sin\left[\frac{\pi}{\omega_1}\left((h(s)+1)\,\omega_3 - v(s)\omega_2\right)\right]}$$
$$= \sum_Y q^{|Y|} \prod_{s \in Y} \frac{1}{\left(y^{v(s)+1} - x^{h(s)}\right)\left(y^{v(s)} - x^{h(s)+1}\right)}, \tag{3.13}$$

with $q = e^{-\frac{2\pi\delta\sigma}{\omega_1}}$, $y = e^{-2\pi i\frac{\omega_2}{\omega_1}}$ and $x = e^{-2\pi i\frac{\omega_3}{\omega_1}}$. Note that all dependence on the coupling has dropped out and the only $\delta\sigma$ dependence is in $q$. In fact $q = e^{-S_I}$, where $S_I$ is the world-line action for the instanton particle with mass $\delta\sigma\mathcal{R}^{-1}$. Hence, $\delta\sigma\mathcal{R}^{-1}$ plays the role of the Coulomb branch scalar and at its origin the instanton particle is massless.

While we won't explicitly demonstrate it here, one can show that $\mathcal{Z}(\delta\sigma|\omega_1; \omega_2, \omega_3) = (q; x, y)_\infty$, where $(q; x, y)_\infty$ is the shifted $q$-factorial [27, 29],

$$(q; x, y)_\infty \equiv \prod_{n=0}^\infty \prod_{m=0}^\infty (1 - q\,x^n y^m), \qquad |x| < 1,\ |y| < 1, \tag{3.14}$$

and its generalization to other regimes for $x$ and $y$. This, and the fact that $S_3(-x; \vec{\omega}) = S_3(x + \Delta; \vec{\omega})$ is enough to show that

$$\exp\left(-\frac{\pi\varrho_5}{3}\left(\delta\sigma^3 - \frac{\omega_1^3 + \omega_2^2 + \omega_3^2 + 3(\omega_1\omega_2 + \omega_2\omega_3 + \omega_3\omega_1)}{2}\delta\sigma\right)\right)$$
$$\times \mathcal{Z}(\delta\sigma|\omega_1; \omega_2, \omega_3)\mathcal{Z}(\delta\sigma|\omega_1; -\omega_2, -\omega_3)\mathcal{Z}(\delta\sigma|\omega_2; \omega_3, \omega_1)\mathcal{Z}(\delta\sigma|\omega_2; -\omega_3, -\omega_1)$$
$$\times \mathcal{Z}(\delta\sigma|\omega_3; \omega_1, \omega_2)\mathcal{Z}(\delta\sigma|\omega_3; -\omega_1, -\omega_2)$$
$$= S_3(\delta\sigma, \vec{\omega})S_3(-\delta\sigma, \vec{\omega}). \tag{3.15}$$

Hence, summing over all instantons reproduces the perturbative contribution to the partition function. Note that the instanton partition function in (3.13) is not invariant under $\delta\sigma \to -\delta\sigma$. However, the complete expression in (3.15) is invariant, which is a consequence of the Weyl symmetry for the instanton $SU(2)$ gauge theory.

## 4 Instantons in seven dimensions

In this section we analyze the negative coupling region of the 7$d$ SYM in light of what we have learnt from the five dimensional case. The first step in this undertaking is to understand what kind of non-perturbative contributions exist on the squashed sphere and which ones are of importance in the considered limit.

## 4.1   The instanton partition function

Inspired by the previous section, we start from considering whereto instantons localize. The first thing to note then is that for generic squashing parameters almost all Reeb orbits do not close but are dense on the $T^4$ over the toric base. This might suggest, as in five dimensions, that nonperturbative configurations should live on the closed orbits over the fixed points on the six-dimensional base. As in the five-dimensional case the space transverse to each closed orbit can be replaced with a twisted $\mathbb{C}^3$. Their contribution is then given by the partition function of the ADHM quiver quantum mechanics associated to the $D0-D6$ brane systems [33–35]. The classical contribution of these configurations is zero. One could introduce the term [11, 24]

$$\frac{\vartheta}{48\pi^3}\int F\wedge F\wedge F\wedge \kappa \tag{4.1}$$

to measure these instantons. However supersymmetrizing this term would lead to a $\text{Tr}\sigma^3$ term in the localized action and thus we choose to not include it. Moreover, the quantum contribution of these instantons is independent of the Coulomb branch parameters. Hence, these instantons contribute an over all $g_{\text{YM}}^2$-independent-factor to the partition function and are not of interest.

In seven dimensions there can be other non-perturbative configurations. A simple scaling argument shows that only co-dimension 4 configurations can contribute to the Yang-Mills action. We are thus led to consider configurations that live on the squashed $\mathbb{S}^3 \subset \mathbb{S}^7$ invariant under the action of the Reeb vector. That is to say, the non-trivial BPS solutions have support on the subspaces

$$|z_i|^2 + |z_j|^2 = 1\,, \quad i \neq j\,, \quad i,j = 1,2,3,4\,. \tag{4.2}$$

We then conjecture that any of the forms in the localization locus in (2.8) that have components on the $\mathbb{S}^3$ are forced to be zero. This sets $\Phi^\pm = \check{F} = 0$. From the fourth equation in (2.8) it follows that $\widehat{F}^+ = 0$. Hence, we find that only contact instantons satisfying

$$*F = \frac{1}{2}F\wedge \kappa \wedge \mathrm{d}\kappa\,, \tag{4.3}$$

supported on a fixed three-sphere are allowed. These describe membranes wrapping the $\mathbb{S}^3$ which are point-like on the four-dimensional space transverse to the $\mathbb{S}^3$. In Appendix B we show how such membranes can be obtained by uplifting point-like instantons from four dimensions.

To determine the instanton contribution to the partition function let us recall how one derives the instanton partition function in (3.5). Instanton solutions on $\mathbb{R}^4$ were classified in [36] in terms of a set of equations for the ADHM variables. Supersymmetrizing and assuming an $\Omega$-background for the $\mathbb{R}^4$, the instantons become point-like [20]. Once the space gets lifted to $\mathbb{R}^4 \times \mathbb{S}^1$, one ends up with a supersymmetric quantum mechanics on the circle for the ADHM variables [37–40]. Due to the twisting on the $\mathbb{R}^4$ this quantum mechanics describes instantons point-like in the directions transverse to the circle. From here the partition function in (3.5) can be computed, where $\omega_2$ and $\omega_3$ play the role of Nekrasov's equivariant parameters $\varepsilon_{1,2}$ and $\frac{2\pi}{\omega_1}$ is the circumference of the circle.

From this brief review of the five-dimensional case, let us give an intuitive argument for the instanton contribution in seven dimensions. A more technical explanation is given in Appendix B. Here we have membrane-like instantons on $\mathbb{R}^4 \times \mathbb{S}^3$, which wrap the $\mathbb{S}^3$ and are point-like on $\mathbb{R}^4$. Of the four squashing parameters on the $\mathbb{S}^7$, two squash the $\mathbb{S}^3$ and the other two twist the $\mathbb{R}^4$. The choice of which parameters do what depends on which fixed three-sphere is being considered. Moreover, on $\mathbb{S}^7$ there are 16 supersymmetries so we expect

some similarity to (3.5) at $\mu = 0$ where the five-dimensional theory is enhanced to $\mathcal{N} = 2$ supersymmetry.

The difference on $\mathbb{S}^7$ is that the ADHM variables are not the fields for a supersymmetric quantum mechanics anymore, but instead the fields for a three-dimensional supersymmetric field theory on the squashed $\mathbb{S}^3$. The $k$-instanton contribution is then given by the partition function of an $U(k)$ gauge theory. These theories can be localized and the partiton function can be written as a matrix integral involving the double sine function $S_2(z; \omega_1, \omega_2)$ [41]. Computing the matrix integrals requires a contour prescription. Using the same prescription as in $4d$ and $5d$, the instanton contribution can then be computed and it involves the same sum over colored partitions as in $4d$ and $5d$.

Hence, the discussion of the last two paragraphs naturally suggests, based on (3.5), that the instanton partition function coming from the squashed three-sphere defined by $|z_1|^2 + |z_2|^2 = 1$ takes the form

$$
Z_{\text{inst}}\left(i\sigma; \omega_1, \omega_2; \omega_3, \omega_4\right) = \sum_{\vec{Y}} e^{-\frac{4\pi^2 |\vec{Y}|}{g_{\text{YM}}^2} \frac{2\pi^2 \mathcal{R}^3}{\omega_1 \omega_2}} \prod_{i,j=1}^{N} \prod_{s \in Y_i} \Bigg\{
$$

$$
\frac{S_2\left(i\sigma_{ji} + \frac{\Delta}{2} - (v_i(s)+1)\omega_3 + h_j(s)\omega_4; \omega_1, \omega_2\right) S_2\left(i\sigma_{ij} + \frac{\Delta}{2} - (h_j(s)+1)\omega_4 + v_i(s)\omega_3; \omega_1, \omega_2\right)}{S_2\left(i\sigma_{ji} - (v_i(s)+1)\omega_3 + h_j(s)\omega_4; \omega_1, \omega_2\right) S_2\left(i\sigma_{ij} - (h_j(s)+1)\omega_4 + v_i(s)\omega_3; \omega_1, \omega_2\right)} \Bigg\},
\tag{4.4}
$$

where now $\Delta = \omega_1 + \omega_2 + \omega_3 + \omega_4 = 4$. Notice that the instantons come weighted with the usual gauge factor $\frac{4\pi^2}{g_{\text{YM}}^2}$, which in seven dimensions has units of a tension, multiplied by the volume of the squashed $\mathbb{S}^3$.

In product form the double sine is given by the expressions

$$
\begin{aligned}
S_2(z; \omega_1, \omega_2) &= \exp\left(+\frac{\pi i}{2} B_{22}(z; \omega_1, \omega_2)\right) \prod_{j \geq 0}^{\infty} \frac{\left(1 - e^{2\pi i\left(\frac{z}{\omega_2} + j\frac{\omega_1}{\omega_2}\right)}\right)}{\left(1 - e^{2\pi i\left(\frac{z}{\omega_1} - (j+1)\frac{\omega_2}{\omega_1}\right)}\right)} \\
&= \exp\left(-\frac{\pi i}{2} B_{22}(z; \omega_1, \omega_2)\right) \prod_{j \geq 0}^{\infty} \frac{\left(1 - e^{-2\pi i\left(\frac{z}{\omega_1} + j\frac{\omega_2}{\omega_1}\right)}\right)}{\left(1 - e^{-2\pi i\left(\frac{z}{\omega_2} - (j+1)\frac{\omega_1}{\omega_2}\right)}\right)},
\end{aligned}
\tag{4.5}
$$

$$
B_{22}(z; \omega_1, \omega_2) = \frac{z^2}{\omega_1 \omega_2} - \frac{\omega_1 + \omega_2}{\omega_1 \omega_2} z + \frac{\omega_1^2 + \omega_2^2 + 3\omega_1 \omega_2}{6\omega_1 \omega_2}.
\tag{4.6}
$$

where we have assumed that $\text{Im}(\omega_1/\omega_2) > 0$. Using this, it is straightforward to establish that

$$
S_2\left(z + \frac{\omega_1 + \omega_2}{2}; \omega_1, \omega_2\right) S_2\left(-z + \frac{\omega_1 + \omega_2}{2}; \omega_1, \omega_2\right) = 1.
\tag{4.7}
$$

This allows us to simplify (4.4) to

$$
\begin{aligned}
Z_{\text{inst}}\left(i\sigma; \omega_1, \omega_2; \omega_3, \omega_4\right) &= \sum_{\vec{Y}} e^{-\frac{4\pi^2 |\vec{Y}|}{g_{\text{YM}}^2} \frac{2\pi^2 \mathcal{R}^3}{\omega_1 \omega_2}} \prod_{i,j=1}^{N} \prod_{s \in Y_i} \Bigg\{ \\
&\frac{1}{S_2\left(i\sigma_{ji} - (v_i(s)+1)\omega_3 + h_j(s)\omega_4; \omega_1, \omega_2\right) S_2\left(i\sigma_{ij} - (h_j(s)+1)\omega_4 + v_i(s)\omega_3; \omega_1, \omega_2\right)} \Bigg\} \\
&= \sum_{\vec{Y}} e^{-\frac{4\pi^2 |\vec{Y}|}{g_{\text{YM}}^2} \frac{2\pi^2 \mathcal{R}^3}{\omega_1 \omega_2}} \prod_{i,j=1}^{N} \prod_{s \in Y_i} \frac{S_2\left(4 + i\sigma_{ij} - (h_j(s)+1)\omega_4 + v_i(s)\omega_3; \omega_1, \omega_2\right)}{S_2\left(i\sigma_{ij} - (h_j(s)+1)\omega_4 + v_i(s)\omega_3; \omega_1, \omega_2\right)}.
\end{aligned}
\tag{4.8}
$$

Note that the original "hypermultiplet" term in (4.4) is equal to 1, which is pleasing since $\mathcal{N} = 1$ super Yang-Mills in seven dimensions only has a vector multiplet.

## 4.2 Instantons at small negative coupling

Let us now see how instantons contribute when the coupling is small but negative. As in the previous section let us specialize to the case of $SU(2)$ and assume that we have a small negative coupling. From the saddle point equation in (2.17) we see that we should set $\sigma_{12} = -\frac{2\pi^3 \mathcal{R}^3}{g_{YM}^2} + \delta\sigma$, where we assume that $|\delta\sigma| \ll -\frac{2\pi^3 \mathcal{R}^3}{g_{YM}^2}$. As before, we have two types of terms, depending on whether or not $i = j$. For a given $\vec{Y} = (Y_1, Y_2)$, the contribution from all terms with $i \neq j$ in $Z_{\text{inst}}(i\sigma; \omega_1, \omega_2; \omega_3, \omega_4)$ is

$$\prod_{s_1 \in Y_1} \frac{S_2\big(4 + i\sigma_{12} - (h_2(s_1) + 1)\omega_4 + v_1(s_1)\omega_3; \omega_1, \omega_2\big)}{S_2\big(i\sigma_{12} - (h_2(s_1) + 1)\omega_4 + v_1(s_1)\omega_3; \omega_1, \omega_2\big)}$$

$$\times \prod_{s_2 \in Y_2} \frac{S_2\big(4 - i\sigma_{12} - (h_1(s_2) + 1)\omega_4 + v_2(s_2)\omega_3; \omega_1, \omega_2\big)}{S_2\big(-i\sigma_{12} - (h_1(s_2) + 1)\omega_4 + v_2(s_2)\omega_3; \omega_1, \omega_2\big)}$$

$$\approx \prod_{s_1 \in Y_1} \exp\left(-\frac{2\pi}{\omega_1 \omega_2}\big(2\sigma_{12} - i(2v_1(s_1) + 1)\omega_3 + i(2h_2(s_1) + 1)\omega_4\big)\right)$$

$$\times \prod_{s_2 \in Y_2} \exp\left(-\frac{2\pi}{\omega_1 \omega_2}\big(2\sigma_{12} + i(2v_2(s_2) + 1)\omega_3 - i(2h_1(s_2) + 1)\omega_4\big)\right)$$

$$= e^{\frac{8\pi^4 \mathcal{R}^3 |\vec{Y}|}{g_{YM}^2 \omega_1 \omega_2}} e^{-\frac{4\pi i |\vec{Y}|}{\omega_1 \omega_2}\delta\sigma} \prod_{s_1 \in Y_1} \exp\left(+\frac{2\pi i}{\omega_1 \omega_2}\big((2v_1(s_1) + 1)\omega_3 + (2h_1(s_1) + 1)\omega_4\big)\right)$$

$$\times \prod_{s_2 \in Y_2} \exp\left(-\frac{2\pi i}{\omega_1 \omega_2}\big((2v_2(s_2) + 1)\omega_3 + (2h_2(s_2) + 1)\omega_4\big)\right), \tag{4.9}$$

where we made use of (A.1) to get the last expression. The terms with $i = j$ are

$$\prod_{s_1 \in Y_1} \frac{S_2\big(4 - (h_1(s_1) + 1)\omega_4 + v_1(s_1)\omega_3; \omega_1, \omega_2\big)}{S_2\big(-(h_1(s_1) + 1)\omega_4 + v_1(s_1)\omega_3; \omega_1, \omega_2\big)}$$

$$\times \prod_{s_2 \in Y_2} \frac{S_2\big(4 - (h_2(s_2) + 1)\omega_4 + v_2(s_2)\omega_3; \omega_1, \omega_2\big)}{S_2\big(-(h_2(s_2) + 1)\omega_4 + v_2(s_2)\omega_3; \omega_1, \omega_2\big)}$$

$$= \prod_{s_1 \in Y_1} \frac{S_2\big(2 - x(s_1) + \frac{\omega_1 + \omega_2}{2}; \omega_1, \omega_2\big)}{S_2\big(-2 + x(s_1) + \frac{\omega_1 + \omega_2}{2}; \omega_1, \omega_2\big)} \prod_{s_2 \in Y_2} \frac{S_2\big(2 - x(s_2) + \frac{\omega_1 + \omega_2}{2}; \omega_1, \omega_2\big)}{S_2\big(-2 + x(s_2) + \frac{\omega_1 + \omega_2}{2}; \omega_1, \omega_2\big)}, \tag{4.10}$$

where we have defined $x(s) \equiv (v(s) + 1/2)\omega_3 - (h(s) + 1/2)\omega_4$. Note that in the last line in (4.10) every factor is invariant under $x(s) \to -x(s)$, and hence under the explicit substitution $\omega_{3,4} \to -\omega_{3,4}$. Therefore, in the negative weak coupling approximation, we can factorize the instanton contribution as

$$Z_{\text{inst}}(i\sigma; \omega_1, \omega_2; \omega_3, \omega_4) \approx \mathcal{Z}(\delta\sigma | \omega_1, \omega_2; \omega_3, \omega_4)\mathcal{Z}(\delta\sigma | \omega_1, \omega_2; -\omega_3, -\omega_4), \tag{4.11}$$

where

$$\mathcal{Z}(\delta\sigma | \omega_1, \omega_2; \omega_3, \omega_4) =$$

$$\sum_Y e^{-\frac{4\pi i |Y|}{\omega_1 \omega_2}\delta\sigma} \prod_{s \in Y} \exp\left(-\frac{2\pi i}{\omega_1 \omega_2}\big((2v(s) + 1)\omega_3 + (2h(s) + 1)\omega_4\big)\right) \tag{4.12}$$

$$\times \frac{S_2\big(2 + x(s) + \frac{\omega_1 + \omega_2}{2}; \omega_1, \omega_2\big)}{S_2\big(-2 + x(s) + \frac{\omega_1 + \omega_2}{2}; \omega_1, \omega_2\big)}.$$

The argument of the exponent in the leading term has the form

$$-\text{vol}(\mathbb{S}^3)|\vec{Y}|\frac{2\,\delta\sigma}{\pi\,\mathcal{R}^3}\,, \tag{4.13}$$

which is the contribution one expects from a membrane with tension $T = \frac{2}{\pi\mathcal{R}^3}\delta\sigma$ wrapping the three-sphere $|\vec{Y}|$ times.

As in the five-dimensional case, the instanton partition function in (4.8) is not an even function of $\delta\sigma$. However, we suspect that when combined with the dominant contribution of the perturbative partition function in (2.27), which is given by

$$\mathcal{Z}_{\text{pert}} \sim \exp\left[\frac{2\pi^4\mathcal{R}^3\varrho}{g_{\text{YM}}^2}\delta\sigma^2 - \frac{\pi}{3}\varrho\left(2\,\delta\sigma^3 - \delta\sigma\sum_{i<j}\omega_i\omega_j\right)\right]\,, \tag{4.14}$$

the full partition function will be invariant under $\delta\sigma \to -\delta\sigma$.

The expression in (4.12) is divergent at the round sphere limit where $\omega_i \to 1$ for all $i$. This is expected since at the round sphere limit there are zero modes, reflecting the enhancement of supersymmetry and hence the freedom to choose new Reeb orbits. However, we also expect the complete partition function to be convergent when summing over the contributions of all three-spheres, since the divergence is really an artifact of the localization. If we consider the one instanton contribution in (4.12), where $x(s) = \frac{1}{2}(\omega_3 - \omega_4)$, we can rewrite the double sines as

$$\frac{S_2\left(2+\frac{\omega_3-\omega_4}{2}+\frac{\omega_1+\omega_2}{2};\omega_1,\omega_2\right)}{S_2\left(-2+\frac{\omega_3-\omega_4}{2}+\frac{\omega_1+\omega_2}{2};\omega_1,\omega_2\right)} = S_2\left(\omega_3+\omega_1+\omega_2;\omega_1,\omega_2\right)S_2\left(\omega_4+\omega_1+\omega_2;\omega_1,\omega_2\right)$$

$$= \frac{S_2\left(\omega_3;\omega_1,\omega_2\right)S_2\left(\omega_4;\omega_1,\omega_2\right)}{16\sin\frac{\pi\omega_3}{\omega_1}\sin\frac{\pi\omega_4}{\omega_1}\sin\frac{\pi\omega_3}{\omega_2}\sin\frac{\pi\omega_4}{\omega_2}}\,. \tag{4.15}$$

In the last expression, as $\omega_i \to 1$ for all $i$, the numerator approaches 1, while the denominator has a fourth order zero. To see that these poles cancel when summing over all six three-spheres we can expand $\omega_3$ about $\omega_1$ and $\omega_4$ about $\omega_2$ and then use the fact that $S_2\left(\omega_1;\omega_1,\omega_2\right)S_2\left(\omega_2;\omega_1,\omega_2\right) = 1$, along with the relations for the derivatives

$$\frac{S_2'(\omega_1;\omega_1,\omega_2)}{S_2(\omega_1;\omega_1,\omega_2)} = -\frac{1}{\sqrt{\omega_1\omega_2}}\left(1-\frac{1}{3}\left(\frac{1}{8}+\frac{\pi^2}{6}\right)\delta^2 + \text{O}(\delta^4)\right)\,,$$

$$\frac{S_2''(\omega_1;\omega_1,\omega_2)}{S_2(\omega_1;\omega_1,\omega_2)} = \frac{\pi^2}{3}\frac{\delta}{\omega_1\omega_2} + \left(\frac{S_2'(\omega_1;\omega_1,\omega_2)}{S_2(\omega_1;\omega_1,\omega_2)}\right)^2\,,$$

$$\frac{S_2'''(\omega_1;\omega_1,\omega_2)}{S_2(\omega_1;\omega_1,\omega_2)} = \frac{1}{\sqrt{(\omega_1\omega_2)^3}}\left(\frac{2\pi^2}{3}+\text{O}(\delta^2)\right)$$
$$+3\frac{S_2''(\omega_1;\omega_1,\omega_2)}{S_2(\omega_1;\omega_1,\omega_2)}\frac{S_2'(\omega_1;\omega_1,\omega_2)}{S_2(\omega_1;\omega_1,\omega_2)}-2\left(\frac{S_2'(\omega_1;\omega_1,\omega_2)}{S_2(\omega_1;\omega_1,\omega_2)}\right)^3\,,$$

$$\frac{S_2''''(\omega_1;\omega_1,\omega_2)}{S_2(\omega_1;\omega_1,\omega_2)} = \frac{4\pi^4}{45}\frac{\delta}{(\omega_1\omega_2)^2}+4\frac{S_2'''(\omega_1;\omega_1,\omega_2)}{S_2(\omega_1;\omega_1,\omega_2)}\frac{S_2'(\omega_1;\omega_1,\omega_2)}{S_2(\omega_1;\omega_1,\omega_2)}$$
$$+3\left(\frac{S_2''(\omega_1;\omega_1,\omega_2)}{S_2(\omega_1;\omega_1,\omega_2)}\right)^2-12\frac{S_2''(\omega_1;\omega_1,\omega_2)}{S_2(\omega_1;\omega_1,\omega_2)}\left(\frac{S_2'(\omega_1;\omega_1,\omega_2)}{S_2(\omega_1;\omega_1,\omega_2)}\right)^2$$
$$+6\left(\frac{S_2'(\omega_1;\omega_1,\omega_2)}{S_2(\omega_1;\omega_1,\omega_2)}\right)^4\,, \tag{4.16}$$

where $\delta = \frac{\omega_1 - \omega_2}{\sqrt{\omega_1 \omega_2}}$. Derivatives for the argument at $\omega_2$ are found by making the substitution $\omega_1 \leftrightarrow \omega_2$ in (4.16). Using this, one can show after a tedious computation that the divergences cancel in the round sphere limit and that the one instanton contribution in (4.11) at negative weak coupling is

$$
\begin{aligned}
Z_{\text{inst}}^{(1)}(i\sigma; 1, 1, 1, 1) \;\approx\; & e^{-4\pi\delta\sigma}\Bigg[ 12 - \frac{785}{72\pi^2} + \frac{11}{12\pi^4} + \left(\frac{11}{3\pi^3} - \frac{677}{18\pi}\right)\delta\sigma \\
& - \left(\frac{106}{3} - \frac{7}{\pi^2}\right)(\delta\sigma)^2 + \frac{20}{3\pi}(\delta\sigma)^3 + \frac{8}{3}(\delta\sigma)^4 \Bigg].
\end{aligned}
\tag{4.17}
$$

## 5 Concluding remarks

The main results of this paper are (4.8) and (4.12). At present we are not able to significantly simplify either expression by carrying out the sum over all instantons, as one can do with the five-dimensional analogs. Nevertheless, we have seen that the instantons for the $SU(2)$ gauge theory behave like membranes wrapped around the squashed $\mathbb{S}^3$ with a non-negative tension $T = \frac{2}{\pi\mathcal{R}^3}\delta\sigma$. This is reminiscent to what happens for the instanton particles in five dimensions. However, in five dimensions we have seen that the $SU(2)$ theory at negative coupling is equivalent to the same theory at positive coupling. This is certainly not the case in seven dimensions. In the remainder of this section we offer a plausible scenario for the seven-dimensional negative coupling regime.

The $\mathcal{R}$ dependence in the tension suggests that $\delta\sigma$ is not part of a vector multiplet. Since a membrane is minimally coupled to a three-form field, the scalar is expected to lie in a multiplet that also contains this field. In seven dimensions the only such multiplet is the $\mathcal{N} = 2$ graviton multiplet [42].[2] This contains the graviton, a three-form field, an $SU(2)$ triplet of vector fields, and a real scalar. Since we wish to place the theory on $\mathbb{S}^7$ we should consider a Euclidean version of this supergravity theory, where also the $SU(2)$ symmetry becomes $SL(2, \mathbb{R})$. To preserve supersymmetry on-shell this requires that the $SL(2, \mathbb{R})$ symmetry be gauged [43–46]. However, since the theory will be localized which requires it be off-shell, we will assume that it is possible to keep the $SL(2, \mathbb{R})$ global symmetry on $\mathbb{S}^7$ and still maintain off-shell supersymmetry.[3] This has the advantage of matching the global symmetry for the usual Yang-Mills theory at positive coupling.

The bosonic part of the Euclidean action for the ungauged graviton multiplet is [43–46]

$$
\begin{aligned}
S_E \;=\; & \frac{1}{2\kappa_7^2}\int d^7x\Bigg[\sqrt{g}\left(-R + \frac{1}{4}e^{\sqrt{\frac{2}{5}}\rho}\eta_{IJ}F^I_{\;\mu\nu}F^{J\mu\nu} + \frac{1}{48}e^{-2\sqrt{\frac{2}{5}}\rho}G_{\mu\nu\kappa\lambda}G^{\mu\nu\kappa\lambda} + \frac{1}{2}\partial_\mu\rho\,\partial^\mu\rho\right) \\
& - \frac{i}{2}\eta_{IJ}\,C\wedge F^I\wedge F^J\Bigg],
\end{aligned}
\tag{5.1}
$$

where

$$
F^I = dA^I, \quad I = 1, 2, 3, \qquad \eta_{IJ} = \text{diag}\{-, +, +\}, \qquad G = dC.
\tag{5.2}
$$

The various effective couplings on $\mathbb{S}^7$ are easily read off from (5.1), with

$$
g_1^2 \sim \frac{(2\kappa_7^2)^{3/5}}{\mathcal{R}^3}e^{-\sqrt{\frac{2}{5}}\rho}, \qquad g_3^2 \sim \frac{\mathcal{R}}{(2\kappa_7^2)^{1/5}}e^{2\sqrt{\frac{2}{5}}\rho}, \qquad g_\rho^2 \sim \frac{2\kappa_7^2}{\mathcal{R}^5},
\tag{5.3}
$$

---

[2]In the supergravity literature minimal supersymmetry in seven dimensions is called $\mathcal{N} = 2$, reflecting the underlying $R$-symmetry.

[3]Progress in localizing supergravity theories has been made in [47–50]. The localization requires that the theory have a boundary where the fluctuations are zero. Since $\mathbb{S}^7$ has no boundary, this suggests that a proper localization will also require the $H_{2,2}/Z_N$ internal space of the supergravity dual described in [8].

where $g_1$ is the coupling for the three $U(1)$ gauge bosons, $g_3$ is the three-form coupling and $g_\rho$ is the scalar coupling. Note that the three-form field $C_{\mu\nu\lambda}$ couples to the three $U(1)$ instanton terms. Normally there are no $U(1)$ instanton solutions but here we are considering the localized action on $\mathbb{S}^7$ where the twist allows for nontrivial point-like solutions [20,51].

By shifting $\rho$ by a constant and absorbing it into $F_{\mu\nu}$ and $G_{\mu\nu\kappa\lambda}$ we can normalize the instantons such that

$$\int F^1 \wedge F^1 = 8\pi^2 (2\kappa_7^2)^{2/5} k, \qquad k \in \mathbb{Z}_+. \tag{5.4}$$

Notice that there is a preferred direction in the $R$-symmetry space, corresponding to the preferred direction taken for the localization Killing spinor. If we assume that one should take another Euclidean rotation, as one does for the scalar field in the super Yang-Mills multiplet, then the tension of $k$ membranes is

$$T = \frac{4\pi^2 e^{\sqrt{\frac{2}{5}}\rho}}{(2\kappa_7^2)^{3/5}} k \sim \frac{k}{g_1^2 \mathcal{R}^3}, \tag{5.5}$$

which we can see directly from the action in (5.1) or by computing the ADM tension for the classical solution sourced by the instanton [52,53].

Equation (5.5) and the previous expression for the tension suggests that we identify

$$\delta\sigma = \frac{2\pi^3 \mathcal{R}^3}{(2\kappa_7^2)^{3/5}} e^{\sqrt{\frac{2}{5}}\rho} \sim \frac{1}{g_1^2}. \tag{5.6}$$

From (5.6) we then have that

$$\partial_\mu \delta\sigma \partial^\mu \delta\sigma = \frac{2}{5}\left(\frac{2\pi^3 \mathcal{R}^3}{(2\kappa_7^2)^{3/5}}\right)^2 e^{2\sqrt{\frac{2}{5}}\rho} \partial_\mu \rho \partial^\mu \rho, \tag{5.7}$$

which leads to the effective coupling for the $\delta\sigma$ field,

$$g_{\delta\sigma}^2 \sim \frac{\mathcal{R}^6}{(2\kappa_7^2)^{6/5}} e^{2\sqrt{\frac{2}{5}}\rho} \frac{2\kappa_7^2}{\mathcal{R}^5} \sim g_3^2. \tag{5.8}$$

If we compare this to (4.14), we see that

$$g_3^2 \sim -\frac{g_{YM}^2}{\mathcal{R}^3}. \tag{5.9}$$

Since we assume that $\delta\sigma \ll -\frac{\mathcal{R}^3}{g_{YM}^2}$, (5.6) and (5.9) imply that $g_3^2 \ll g_1^2$. Furthermore, we can write $g_3^2 \sim \frac{2\kappa_7^2}{\mathcal{R}^5} g_1^{-4}$, so we must also choose $g_1^2 < 1$ if the three-form coupling is to be stronger than gravity. If we write $g_1^2 \sim \left(\frac{2\kappa_7^2}{\mathcal{R}^5}\right)^\alpha$, then the couplings satisfy

$$\frac{2\kappa_7^2}{\mathcal{R}^5} \ll g_3^2 \ll g_1^2 < 1, \tag{5.10}$$

if $0 < \alpha < 1/3$.

In the region where $-\frac{g_{YM}^2}{\mathcal{R}^3} \ll 1$, we can approximate the partition function as

$$\mathcal{Z} \approx \int d\delta\sigma \, e^{\frac{2\pi^4 \mathcal{R}^3}{g_{YM}^2}\delta\sigma^2} Z_q(\delta\sigma), \tag{5.11}$$

where $Z_q(\delta\sigma)$ contains the coupling independent terms in (4.14) and the contribution of the instantons in (4.12). If we know $Z_q(\delta\sigma)$ we can solve (5.11) by saddle point and find $\delta\sigma$. At present we do not know the behavior of $Z_q(\delta\sigma)$, but if it had the same behavior as in the five-dimensional case with $Z_q(\delta\sigma) \sim \delta\sigma^2$ for $|\delta\sigma| \ll 1$, then at the saddle point $\delta\sigma \sim \sqrt{-\frac{g_{YM}^2}{\mathcal{R}^3}}$. This would correspond to having $\alpha = 1/4$, which is inside the desired window. It would be interesting to explore this further.

## Acknowledgements

We thank L. Cassia, A. Dabholkar, J. Gomis, P. Jefferson, M. Kim, S. Murthy, V. Rodriguez and M. Zabzine for helpful conversations and correspondence. This research was supported in part by Vetenskapsrådet under grants #2016-03503 and #2020-03339, by the Knut and Alice Wallenberg Foundation under grant Dnr KAW 2015.0083, and by the National Science Foundation under Grant No. NSF PHY-1748958. JAM thanks the KITP for hospitality during the course of this work.

## A  An identity

In this section of the appendix we show that

$$\sum_{s_1 \in Y_1}(2h_2(s_1)+1) - \sum_{s_2 \in Y_2}(2h_1(s_2)+1) = \sum_{s_2 \in Y_2}(2h_2(s_2)+1) - \sum_{s_1 \in Y_1}(2h_1(s_1)+1), \quad \text{(A.1)}$$

where $Y_1$ and $Y_2$ are two Young diagrams, $s_1$ and $s_2$ the respective boxes in the diagrams, and $h_i(s)$ is the horizontal distance to the edge of diagram $Y_1$ from box $s$. Let the rows for $Y_1$ be given by $\{\lambda_1, \lambda_2, \ldots, \lambda_n\}$ with $\lambda_i \geq \lambda_{i+1}$. Likewise let the rows for $Y_2$ be $\{\lambda'_1, \lambda'_2, \ldots, \lambda'_{n'}\}$ with $\lambda'_i \geq \lambda'_{i+1}$.

We then have that

$$\sum_{s_1 \in Y_1}(2h_2(s_1)+1) = \sum_{k=1}^{n}\sum_{j=1}^{\lambda_k}\left(2(\lambda'_k - j)+1\right) = \sum_{k=1}^{n}(2\lambda'_k\lambda_k - \lambda_k{}^2), \quad \text{(A.2)}$$

where we use that $\lambda'_k = 0$ if $k > n'$. Likewise, we have that

$$\sum_{s_2 \in Y_2}(2h_1(s_2)+1) = \sum_{k'=1}^{n'}(2\lambda_{k'}\lambda'_{k'} - \lambda'_{k'}{}^2). \quad \text{(A.3)}$$

Hence,

$$\sum_{s_1 \in Y_1}(2h_2(s_1)+1) - \sum_{s_2 \in Y_2}(2h_1(s_2)+1) = \sum_{k'=1}^{n'}\lambda'_{k'}{}^2 - \sum_{k=1}^{n}\lambda_k{}^2. \quad \text{(A.4)}$$

Next, we have that

$$\sum_{s_2 \in Y_2}(2h_2(s_2)+1) = \sum_{k'=1}^{n'}\sum_{j'=1}^{\lambda'_{k'}}\left(2(\lambda'_{k'} - j')+1\right) = \sum_{k'=1}^{n'}\lambda'_{k'}{}^2,$$

$$\sum_{s_1 \in Y_1}(2h_1(s_1)+1) = \sum_{k=1}^{n}\sum_{j=1}^{\lambda_k}\left(2(\lambda'_k - j)+1\right) = \sum_{k=1}^{n}\lambda_k{}^2. \quad \text{(A.5)}$$

Hence, (A.1) is true.

# B  Technicalities regarding instanton contributions

In this part of the appendix we give additional details about the instanton contributions discussed in the main text. We will not attempt to find the full set of solutions to the BPS equations (2.3).

We start by assuming the vanishing of the three-form $\Phi$. Then the only non-trivial component of the gauge field strength is $\hat{F}^-$. Thus, we look for solutions to the equation

$$*F = \frac{1}{2}F \wedge \kappa \wedge d\kappa. \tag{B.1}$$

A set of solutions can be found by uplifting point-like instantons from four dimensions by wrapping them on a unit $\mathbb{S}^3 \subset \mathbb{S}^7$. Treating $\mathbb{S}^7$ as an $\mathbb{S}^3$ fibered over $\mathbb{S}^4$, it is clear that the gauge field configuration for a point-like instanton on the base can be lifted to the $\mathbb{S}^7$ by taking it to be constant along the fiber. The field-strength of the lift will be non-zero only on a single $\mathbb{S}^3$ fiber and will only have components that are transverse to that fiber. By squashing the $\mathbb{S}^7$, the three-spheres the instantons can wrap are the six invariant under the action of the Reeb vector.

It is possible to explicitly check that these uplifted contact instantons satisfy the equation. We choose coordinates $(\theta, \phi, \chi, x^i)$ for $i = 5, 6, 7, 8$, on the $\mathbb{S}^7$ and embed it into $\mathbb{R}^8$ by setting

$$\begin{aligned}
x^1 &= \rho_1 \cos\phi = \sqrt{1-r^2}\cos(\theta)\cos(\phi), \\
x^2 &= \rho_1 \sin\phi = \sqrt{1-r^2}\cos(\theta)\sin(\phi), \\
x^3 &= \rho_2 \cos\chi = \sqrt{1-r^2}\sin(\theta)\cos(\chi), \\
x^4 &= \rho_2 \sin\chi = \sqrt{1-r^2}\sin(\theta)\sin(\chi), \\
1 &\geq r^2 = \rho_3^2 + \rho_4^2 = (x^5)^2 + (x^6)^2 + (x^7)^2 + (x^8)^2.
\end{aligned}$$

This choice explicitly distinguishes between the $\mathbb{S}^3$ coordinates $(\theta, \phi, \chi)$, and the transverse space $(x^5, x^6, x^7, x^8)$. In these coordinates the Reeb vector for the squashed sphere takes the form

$$R^\mu \partial_\mu = \omega_1 \partial_\phi + \omega_2 \partial_\chi + \omega_3 \left(x^5 \partial_{x^6} - x^6 \partial_{x^5}\right) + \omega_4 \left(x^7 \partial_{x^8} - x^8 \partial_{x^7}\right). \tag{B.2}$$

This form of the Reeb vector makes it clear that the supersymmetry approaches that of $S^3_{\frac{\omega_1}{\omega_2}} \times \mathbb{R}^4_{\omega_3, \omega_4}$, that is a squashed three-sphere times the $\Omega$-background, as we go to $x^5 = x^6 = x^7 = x^8 = 0$. For the metric on the squashed sphere we take

$$ds^2 = g_{\mu\nu}dx^\mu dx^\nu = \frac{\alpha^2}{\beta^2}\sum_{i=1}^4 \left(d\rho_i^2 + \rho_i^2 d\phi_i^2\right) + \frac{1}{\beta^2}\left(\sum_i a_i \rho_i^2 d\phi_i\right)^2, \tag{B.3}$$

where $\alpha^2 = 1 - \sum_i a_i^2 \rho_i^2$ and $\beta = 1 + \sum_i a_i \rho_i^2$. This metric is conformally equivalent to the metric in (2.2) and ensures that the Reeb vector has unit norm. With this metric, we have the contact-metric structure [54–56] on the squashed sphere,

$$v = R, \qquad \kappa = g(v, \cdot) = \frac{\alpha^2}{\beta^2}\sum_i \rho_i^2 d\phi_i + \frac{1}{\beta}\sum_i a_i \rho_i^2 d\phi, \qquad g_{\mu\lambda}J^\lambda{}_\nu = d\kappa_{\mu\nu}. \tag{B.4}$$

As we zoom in on the chosen $\mathbb{S}^3$, by taking the limit $r \to 0$ the metric becomes

$$\begin{aligned}
ds^2 = &\frac{1 - a_1^2 \cos^2\theta - a_2^2 \sin^2\theta}{\left(1 + a_1 \cos^2\theta + a_2 \sin^2\theta\right)^2}\left(d\theta^2 + \cos^2\theta d\phi^2 + \sin^2\theta d\chi^2\right) \\
&+ \frac{\left(a_1 \cos^2\theta d\phi + a_2 \sin^2\theta d\chi\right)^2}{\left(1 + a_1 \cos^2\theta + a_2 \sin^2\theta\right)} + \delta_{ij}dx^i dx^j.
\end{aligned} \tag{B.5}$$

On this same $\mathbb{S}^3$ the lifted four-dimensional instantons have a field-strength which satisfies

$$F = -\star_{\mathbb{R}^4} F\,, \tag{B.6}$$

i.e., it has no component along $\mathbb{S}^3$, while on the transverse $\mathbb{R}^4$ it satisfies the usual anti-self-duality condition. It is then straightforward to show that this field strength satisfies the seven-dimensional contact instanton equation,

$$\frac{1}{2}F \wedge k \wedge dk = -\frac{1}{2}F \wedge \mathrm{vol}_{\mathbb{S}^3} = \frac{1}{2}\sqrt{g}\,\varepsilon_{\theta\phi\chi k\ell}{}^{ij}F_{ij}d\theta \wedge d\phi \wedge d\chi \wedge dx^i \wedge dx^j = \star F\,. \tag{B.7}$$

We have thus argued that at least some of the seven-dimensional contact instantons wrap these six distinguished $\mathbb{S}^3$'s in the $\mathbb{S}^7$. Moreover, we noted that the form of the Reeb vector suggests that the supersymmetry approaches that of twisted $\mathbb{R}^4$ times squashed $\mathbb{S}^3$ as we approach these loci. This leads than to the following two conjectures:

1. On the squashed seven-sphere all contact membrane instantons localize to the six distinguished three-spheres.

2. The instanton contribution of each such three-sphere can be computed from ADHM data on this squashed $\mathbb{S}^3$.

In the rest of this section we build upon these conjectures to derive two formulas. First the ADHM data on the $\mathbb{S}^3$ will give us an integral formula for the instantons,

$$\left(\frac{S_2(-\omega_3-\omega_4)}{S_2(-\omega_3)S_2(-\omega_4)}\right)^k \int \frac{d^k\phi}{k!\left(i\sqrt{\omega_1\omega_2}\right)^k} \prod_{i=1}^{K}\prod_{A=1}^{N} \frac{1}{S_2(\phi_i-a_A)S_2(-\phi_i+a_A-(\omega_3+\omega_4))}$$
$$\prod_{i\neq j}\frac{S_2(\phi_{ij})S_2(\phi_{ij}-\omega_3-\omega_4)}{S_2(\phi_{ij}-\omega_3)S_2(\phi_{ij}-\omega_4)}\,. \tag{B.8}$$

Note that here and in the rest of this appendix we will be suppressing the two squashing parameters $\omega_1, \omega_2$ for the three-sphere when we write the double sine function. The unusual normalization factor for the contour integrals is because $S_2'(0) = \frac{2\pi}{\sqrt{\omega_1\omega_2}}$. In the second step we evaluate this integral by giving a prescription for which poles should be enclosed by the integration contour, giving us the $k$-instanton result

$$Z_k = \sum_{|Y|=k}\prod_{i,j=1}^{N}\prod_{s\in Y_i}\frac{1}{S_2\left(i\sigma_{ji}-(v_i(s)+1)\,\omega_3+h_j(s)\omega_4\right)S_2\left(i\sigma_{ij}-(h_j(s)+1)\,\omega_4+v_i(s)\omega_3\right)}\,. \tag{B.9}$$

## B.1 ADHM construction

The ADHM data consists of two adjoint chiral multiplets $B_{1,2}$ as well as two chirals $I, J$ respectively in the representations $(\overline{N}, k)$ and $(N, \overline{k})$ of $U(N) \times U(k)$. They are subject to the constraints

$$\mu_{\mathbb{R}} \equiv [B_1, B_1^\dagger] + [B_2, B_2^\dagger] + II^\dagger - J^\dagger J = 0\,,$$
$$\mu_{\mathbb{C}} \equiv [B_1, B_2] + IJ = 0\,. \tag{B.10}$$

This ADHM data parametrizes the instanton moduli space. Regularization of this moduli space requires modifying the constraints to $\vec{s} = (\mu_{\mathbb{R}}-\zeta, \mu_{\mathbb{C}}) = 0$. In addition, we have the $U(k)$ vector

multiplet $(A, \psi, \phi, \alpha, D)$. Twisted supersymmetry of the vector and chiral multiplets is

$$
\begin{aligned}
\mathcal{Q}A &= \psi, & \mathcal{Q}\psi &= \iota_R \mathcal{L}_R + [\phi, A], \\
\mathcal{Q}\phi &= 0, & & \\
\mathcal{Q}\alpha &= D, & \mathcal{Q}D &= \iota_R d\alpha + [\phi, \alpha], \\
\mathcal{Q}B_{1,2} &= \psi_{1,2}, & \mathcal{Q}\psi_{1,2} &= \iota_R dB_{1,2} + [\phi, B_{1,2}] + \epsilon_{1,2} B_{1,2}, \\
\mathcal{Q}\chi_{1,2} &= Y_{1,2}, & \mathcal{Q}Y_{1,2} &= \iota_R d\chi_{1,2} + [\phi, \chi_{1,2}] - \epsilon_{1,2} \chi_{1,2}, \\
\mathcal{Q}I &= \psi_I, & \mathcal{Q}\psi_I &= \iota_R dI + \phi I - I a, \\
\mathcal{Q}\chi_I &= Y_I, & \mathcal{Q}Y_I &= \iota_R d\chi_I + \phi \chi_I - \chi_I a, \\
\mathcal{Q}J &= \psi_J, & \mathcal{Q}\psi_J &= \iota_R dJ - J\phi + aJ - (\epsilon_1 + \epsilon_2)J, \\
\mathcal{Q}\chi_J &= Y_J, & \mathcal{Q}Y_J &= \iota_R d\chi_J - \chi_J \phi + a\chi_J - (\epsilon_1 + \epsilon_2)\chi_J,
\end{aligned}
\tag{B.11}
$$

where we have used the cohomological fields in [57, 58]. In addition, we need a projection multiplet to lift from $\mu_{\mathbb{R}}^{-1}(\zeta) \cap \mu_{\mathbb{C}}^{-1}(0)/U(k)$ to $\mu_{\mathbb{R}}^{-1}(\zeta) \cap \mu_{\mathbb{C}}^{-1}(0)$, and Fadeev-Popov ghosts to gauge fix. We do not focus on these latter points here. Instead we concentrate on the effect of the ADHM constraints as this is the only thing that is nonstandard for three-dimensional localization. To impose them we could simply introduce a delta function $\delta(\vec{s})$ into the path integral. As is done fort gauge fixing, the delta function can be replaced by an insertion of the Gaussian factor

$$
\exp\left(-\int d^3 x \frac{1}{2g_H} \text{Tr}(s_{\mathbb{R}}^2 + |s_{\mathbb{C}}|^2)\right)
$$
$$
= \int DH_{\mathbb{R}} DH_{\mathbb{C}} \exp\left(-\int d^3 x \left(\frac{g_H}{2} \text{Tr}(H_{\mathbb{R}}^2 + |H_{\mathbb{C}}|^2) + \text{i} \, \text{Tr}(H_{\mathbb{R}} s_{\mathbb{R}} + H_{\mathbb{C}}^{\dagger} s_{\mathbb{C}})\right)\right). \tag{B.12}
$$

Obviously we should do this so that supersymmetry is preserved. However, it is straightforward to deduce which fields to include to make it supersymmetric. Namely, the density in (B.12) should be part of a positive definite Lagrangian density of the form

$$
\mathcal{Q} \, \text{Tr}\left(\frac{g_H}{2}(\chi_{\mathbb{R}} \mathcal{Q}\chi_{\mathbb{R}} + \chi_{\mathbb{C}}^{\dagger} \mathcal{Q}\chi_{\mathbb{C}}) + \text{i}(\chi_{\mathbb{R}} s_{\mathbb{R}} + \chi_{\mathbb{C}}^{\dagger} s_{\mathbb{C}})\right). \tag{B.13}
$$

It is then clear that we should require the following SUSY transformations,

$$
\begin{aligned}
\mathcal{Q}\chi_{\mathbb{R}} &= H_{\mathbb{R}}, & \mathcal{Q}H_{\mathbb{R}} &= \iota_R d\chi_{\mathbb{R}} + [\phi, \chi_{\mathbb{R}}], \\
\mathcal{Q}\chi_{\mathbb{C}} &= H_{\mathbb{C}}, & \mathcal{Q}H_{\mathbb{C}} &= \iota_R d\chi_{\mathbb{C}} + [\phi, \chi_{\mathbb{C}}] + (\epsilon_1 + \epsilon_2)\chi_{\mathbb{C}}.
\end{aligned}
\tag{B.14}
$$

This way we make sure that $\mathcal{Q}^2 = \mathcal{L}_R + G_{\phi} + G_{\epsilon_1, \epsilon_2}$ squares to the sum of bosonic symmetries. Note that $\chi_{\mathcal{C}}, H_{\mathcal{C}}$ transform under the torus action on the $\mathbb{R}^4$ such that $H_{\mathbb{C}}^{\dagger} s_{\mathbb{C}}$ is invariant. Diagonalizing $\mathcal{Q}^2$, it is easy to see that the $(\chi_{\mathbb{R}}, H_{\mathbb{R}})$ multiplet does not contribute to the partition function, after taking care of its zero mode.

The other ghost multiplet needs more work, as we have to extend the multiplet to $(\chi_{\mathbb{C}}, H_{\mathbb{C}}, \xi_{\mathbb{C}}, D_{\mathbb{C}})$. This can be seen by remembering that so far we are using a twisted version of off-shell $\mathcal{N} = 2$ supersymmetry. Then the field $\chi_{\mathbb{C}}$ is still a fermionic scalar in the non-twisted formalism, while $H_{\mathbb{C}}$ comes from a bosonic spinor $H^{\alpha}$. Twisting this spinor gives the desired $H_{\mathbb{C}}$ as well as the auxiliary $D_{\mathbb{C}}$. Matching the number of bosonic and fermionic degrees of freedom in the non-twisted formalism introduces the fermionic auxiliary scalar $\xi$, which becomes $\xi_{\mathbb{C}}$ after twisting. Schematically, the SUSY transformation rules of the multiplet $(\chi_{\mathbb{C}}, H^{\alpha}, \xi)$ are

$$
\begin{aligned}
\{Q^{\alpha}, \chi_{\mathbb{C}}\} &= H^{\alpha}, \\
[Q^{\alpha}, H^{\beta}] &= (\gamma^{\mu})^{\alpha\beta} \partial_{\mu}\chi + \epsilon^{\alpha\beta}\xi, \\
\{Q^{\alpha}, \xi\} &= (\gamma^{\mu})^{\alpha\beta} D_{\mu} H_{\beta}.
\end{aligned}
\tag{B.15}
$$

We observe that this multiplet is a ghost version of a chiral multiplet, *viz.* with the statistics reversed. Consequently this multiplet will contribute the inverse of a corresponding chiral multiplet to the partition function. Hence, the respective contributions to (B.8) are

$$
\begin{aligned}
(A, \psi, \phi, \alpha, D) &\rightarrow & S_2(\phi_{ij}), \\
(\chi_{\mathbb{C}}, H_{\mathbb{C}}, \xi_{\mathbb{C}}, D_{\mathbb{C}}) &\rightarrow & S_2(\phi_{ij} - \omega_3 - \omega_4), \\
(B_1, \psi_1, \chi_1, Y_1) &\rightarrow & S_2(\phi_{ij} - \omega_3)^{-1}, \\
(B_2, \psi_2, \chi_2, Y_2) &\rightarrow & S_2(\phi_{ij} - \omega_4)^{-1}, \\
(I, \psi_I, \chi_I, Y_I) &\rightarrow & S_2(\phi_i - a_A)^{-1}, \\
(J, \psi_J, \chi_J, Y_J) &\rightarrow & S_2(-\phi_i + a_A - (\omega_3 + \omega_4))^{-1}.
\end{aligned}
\tag{B.16}
$$

## B.2 Towards $k$-instantons in the abelian theory

In this section we sketch an argument that provides more evidence for the general validity of our conjecture. Here we focus on an abelian gauge group and look at the $k$-instanton contribution coming from a particular Young diagram, $Y$. From here we can find a $k + 1$-instanton contribution by adding a box to $Y$ at an appropriate place to obtain a new Young diagram, $Y_+$. We now show that $\frac{Z_{k+1,Y+}}{Z_{k,Y}}$ computed from our conjecture matches the ADHM contour formula.

We label the position of each box $s$ in the Young diagram by a pair of integers $(n, m)$. We add a box at a position $(\hat{n}, \hat{m})$ to the diagram $Y$ to obtain $Y_+$. The horizontal and vertical distances to the edge for boxes in $Y_+$ are related to those in $Y$ as follows:

$$
h_{Y_+}(s) = \begin{cases} h_Y(s), & m \neq \hat{m}, \\ h_Y(s) + 1 = \hat{n} - n, & m = \hat{m}, \end{cases}
\tag{B.17}
$$

and

$$
v_{Y_+}(s) = \begin{cases} v_Y(s), & n \neq \hat{n}, \\ v_Y(s) + 1 = \hat{m} - m, & n = \hat{n}, \end{cases}
\tag{B.18}
$$

with

$$
h_{Y_+}(\hat{n}, \hat{m}) = v_{Y_+}(\hat{n}, \hat{m}) = 0.
\tag{B.19}
$$

Using this and our conjecture we can write the ratio as

$$
\begin{aligned}
\frac{Z_{k+1,Y+}}{Z_{k,Y}} = {} & \frac{1}{S_2(-\omega_3) S_2(-\omega_4)} \\
& \times \prod_{n=1}^{\hat{n}-1} \frac{S_2\left(-\left(Y_n^{\mathrm{T}} - \hat{m} + 1\right)\omega_3 + (\hat{n} - n - 1)\omega_4\right) S_2\left(-(\hat{n} - n)\omega_4 + \left(Y_n^{\mathrm{T}} - \hat{m}\right)\omega_3\right)}{S_2\left(-\left(Y_n^{\mathrm{T}} - \hat{m} + 1\right)\omega_3 + (\hat{n} - n)\omega_4\right) S_2\left(-(\hat{n} - n + 1)\omega_4 + \left(Y_n^{\mathrm{T}} - \hat{m}\right)\omega_3\right)} \\
& \times \prod_{m=1}^{\hat{m}-1} \frac{S_2\left(-(\hat{m} - m)\omega_3 + (Y_m - \hat{n})\omega_4\right) S_2\left(-(Y_m - \hat{n} + 1)\omega_4 + (\hat{m} - m - 1)\omega_3\right)}{S_2(-(\hat{m} - m + 1)\omega_3 + (Y_m - \hat{n})\omega_4) S_2\left(-(Y_m - \hat{n} + 1)\omega_4 + (\hat{m} - m)\omega_3\right)},
\end{aligned}
\tag{B.20}
$$

where $Y_m$ is the number of boxes in row $m$ of $Y$ and $Y_n^T$ is the number of boxes in column $n$ of $Y$. The first term in (B.20) comes from the square $(\hat{n}, \hat{m})$, the first product comes from the squares with $m = \hat{m}$, and the second product comes from the squares with $n = \hat{n}$.

Many terms in the products in (B.20) cancel. To see the cancellations we can visualize the products in a diagram, as shown in figure 2a. The key thing to note is that the boxes $\left(n, Y_n^{\mathrm{T}}\right)$

and $(Y_m, m)$ are on the edge of the diagram and the product over $n$ moves us along the bottom edges from the left until $n = \hat{n} - 1$ and the product over $m$ moves us along the side edges from the top until $m = \hat{m} - 1$. We can then represent each term in the product of (B.20) as four boxes, with each box representing an $S_2$ in the product. If we define the position of a box at $(n, m)$ as $n\omega_4 + m\omega_3$ and the distance between two boxes as the difference in their positions, then the arguments of the $S_2$ are plus or minus the distances between boxes in $Y$ and the new box at $(\hat{n}, \hat{m})$. The boxes are further labeled by a '+' or a '−', indicating if the $S_2$ is in the numerator or denominator. As we move along the edges most poles and residues cancel between adjacent values of $n$ and $m$.

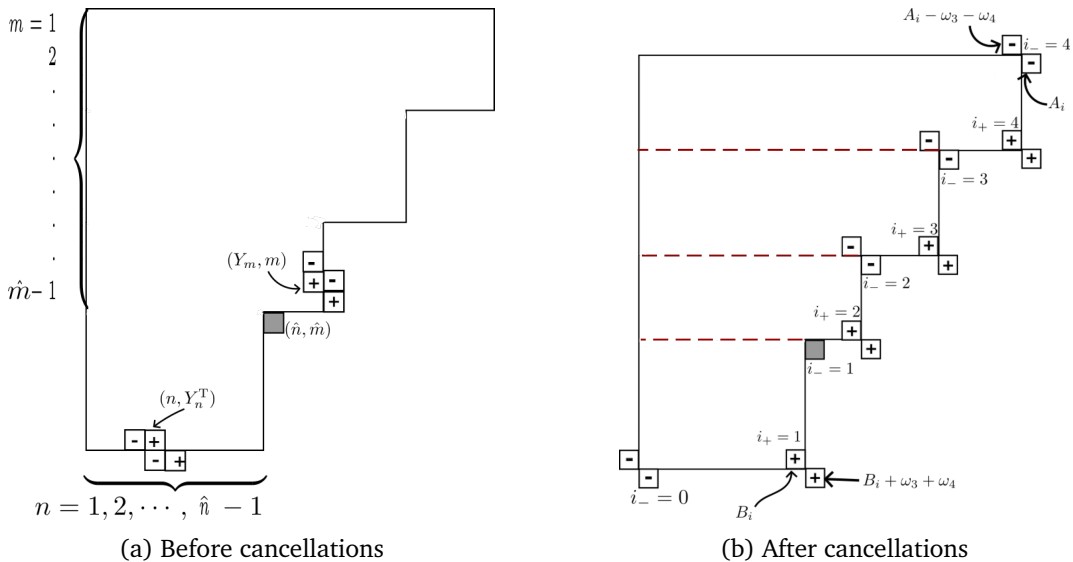

(a) Before cancellations   (b) After cancellations

Figure 2: The left figure is a diagrammatic representation of eq. (B.20). Each term in the products in (B.20) are represented by four boxes along the edge of $Y$, with a '+' box representing an $S_2$ in the numerator and a '−' box representing an $S_2$ in the denominator. After various cancellations the leftover factors in eq. (B.21) can be expressed in terms of the distance of the boxes from the extra box as shown in the right figure. The red dashed lines indicate the division into $n_r$ rectangles.

After cancellations we are left with $2n_r$ boxes contributing to the denominator and $2n_r$ to the numerator, where $n_r$ is the number of stacked rectangles that make up $Y$, as shown in figure 2b. For $i = 0, \cdots n_r$, let $A_i$ be the positions of all allowed boxes that can be added to $Y$. We assume that $i = x$ represents the position of the box that makes $Y_+$. Then the positions of the '−' boxes are at $A_i$ and $A_i - \omega_3 - \omega_4$, $i \neq x$. If we also let $B_i$, $i = 1, \ldots 4$, be the position of the bottom right corner of each rectangle, then the positions of the '+' boxes are located at $B_i$ and $B_i + \omega_3 + \omega_4$. With these definitions we can write (B.20) as

$$\frac{Z_{k+1,Y+}}{Z_{k,Y}} = \frac{1}{S_2(-\omega_3)S_2(-\omega_4)} \frac{\displaystyle\prod_{i=1}^{n_r} S_2(A_x - B_i - \omega_3 - \omega_4)S_2(B_i - A_x)}{\displaystyle\prod_{\substack{i=0 \\ i\neq x}}^{n_r} S_2(A_x - A_i)S_2(A_i - A_x - \omega_3 - \omega_4)}. \tag{B.21}$$

We now show that the ADHM contour integral in (B.8) gives the recursion relation in (B.21). We start by assuming that $Z_{k,Y}$ follows from (B.8), where the contours are chosen so that $\phi_i$ has a pole at $\hat{\phi}_i = a_1 + C_i - \omega_3 - \omega_4$, where $C_i$ is the position of one of the $k$ boxes in

$Y$. It then follows that

$$\frac{Z_{k+1,Y+}}{Z_{k,Y}} = \frac{S_2\left(-\omega_3-\omega_4\right)}{S_2\left(-\omega_3\right)S_2\left(-\omega_4\right)} \int \frac{d\phi_{k+1}}{\mathrm{i}\sqrt{\omega_1\omega_2}} \frac{1}{S_2\left(\phi_{k+1}-a_1\right)S_2\left(-\phi_{k+1}+a_1-\omega_3-\omega_4\right)}$$
$$\times \prod_{i=1}^{k}\prod_{\eta=\pm1} \frac{S_2\left(\eta(\phi_{k+1}-\hat{\phi}_i)\right)S_2\left(\eta(\phi_{k+1}-\hat{\phi}_i)-\omega_3-\omega_4\right)}{S_2\left(\eta(\phi_{k+1}-\hat{\phi}_i)-\omega_3\right)S_2\left(\eta(\phi_{k+1}-\hat{\phi}_i)-\omega_4\right)}. \quad \text{(B.22)}$$

Each term in the product involves eight terms. There are many cancellations which happen after performing the product. A diagrammatic approach allows us to track all cancellations. This is shown in figure 3. Note the resemblance to figure 2b. Thus the final expression takes the form

$$\frac{Z_{k+1,Y+}}{Z_{k,Y}} = \frac{S_2\left(-\omega_3-\omega_4\right)}{S_2\left(-\omega_3\right)S_2\left(-\omega_4\right)} \int \frac{d\phi_{k+1}}{2\pi\mathrm{i}} \frac{\prod_{i=1}^{n_r}S_2\left(\phi_{k+1}-B_i-\omega_3-\omega_4\right)S_2\left(B_i-\phi_{k+1}\right)}{\prod_{i=0}^{n_r}S_2\left(\phi_{k+1}-A_i\right)S_2\left(A_i-\phi_{k+1}-\omega_3-\omega_4\right)}. \quad \text{(B.23)}$$

We now perform the contour integral over $\phi$ picking up the residue associated with the additional box corresponding to $Y_+$. If $A_x$ is the position of the additional box then it is easy to see that we get the ratio (B.21).

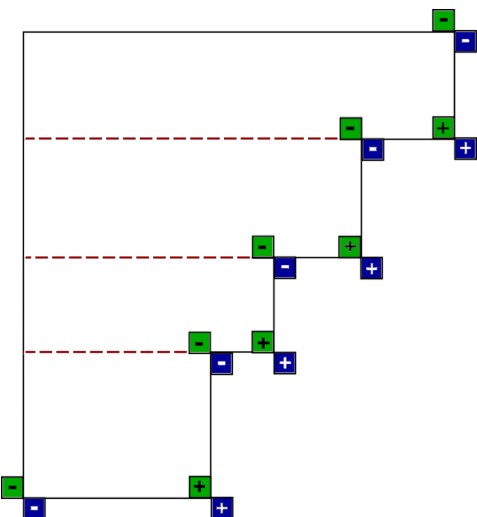

Figure 3: Representation of the integrand for general $k$. We divide the diagram into rectangles as indicated by the dashed red lines. Factors in the numerator correspond to a box at the bottom right corner of each rectangle and a box shifted by $\omega_3 + \omega_4$. Factors in the denominator correspond to each allowed box that can be added to get a valid $k+1$-instanton diagram and a box shifted by $-\omega_3-\omega_4$. We highlight in blue factors with $\phi_{k+1}$, while those in green have $-\phi_{k+1}$.

### B.3 Towards $k$-instantons in the non-abelian theory

In this subsection we extend the argument from the previous subsection to $SU(N)$ super Yang-Mills.

We start with a vector of $N$ Young diagrams $\vec{Y} = (Y_1,\ldots,Y_N)$ with $|\vec{Y}| = k$. Now, $\vec{Y}_+$ is the set of diagrams where we have added to the diagram $Y_{\hat{a}}$ a box at position $(\hat{n},\hat{m})$. The changes

for the horizontal and vertical distances are

$$
h_{Y_{+a}}(s) = \begin{cases} h_{Y_a}(s) + 1 = \hat{n} - n, & m = \hat{m} \text{ and } a = \hat{a}, \\ h_{Y_a}(s), & \text{otherwise}, \end{cases} \tag{B.24}
$$

$$
v_{Y_{+b}}(s) = \begin{cases} v_{Y_b}(s) + 1 = \hat{m} - m, & n = \hat{n} \text{ and } b = \hat{a}, \\ v_{Y_b}(s), & \text{otherwise}, \end{cases} \tag{B.25}
$$

$$
h_{Y_{+\hat{a}}}(\hat{n}, \hat{m}) = v_{Y_{+\hat{a}}}(\hat{n}, \hat{m}) = 0. \tag{B.26}
$$

Starting from our conjectured expression (B.9), the first thing to note is that we only have to keep terms with either index $i$ or $j$ equal to $\hat{a}$. Also, plugging in expressions for the horizontal and vertical distances, it is clear that only the contributions from boxes in row $\hat{m}$ or column $\hat{n}$ do not cancel. With these simplifications we find

$$
\frac{Z_{\vec{Y}_+}}{Z_{\vec{Y}}} = \left( \prod_{b \neq \hat{a}}^{N} \frac{1}{S_2(i\sigma_{b,\hat{a}} - \omega_3 + (Y_{b,\hat{m}} - \hat{n})\omega_4) S_2(i\sigma_{\hat{a},b} - (Y_{b,\hat{m}} - \hat{n} + 1)\omega_4)} \right. \tag{B.27}
$$

$$
\left. \prod_{m=1}^{\hat{m}-1} \frac{S_2\left( i\sigma_{b,\hat{a}} - (\hat{m}-m)\omega_3 + (Y_{b,m} - \hat{n})\omega_4 \right) S_2\left( i\sigma_{\hat{a},b} - (Y_{b,m} - \hat{n} + 1)\omega_4 + (\hat{m}-m-1)\omega_3 \right)}{S_2\left( i\sigma_{b,\hat{a}} - (\hat{m}-m+1)\omega_3 + (Y_{b,m} - \hat{n})\omega_4 \right) S_2\left( i\sigma_{\hat{a},b} - (Y_{b,m} - \hat{n} + 1)\omega_4 + (\hat{m}-m)\omega_3 \right)} \right)
$$

$$
\left( \prod_{a \neq \hat{a}}^{N} \prod_{n=1}^{Y_{a,\hat{m}}} \frac{S_2\left( i\sigma_{\hat{a},a} - (Y_{a,n}^T - \hat{m} + 1)\omega_3 + (\hat{n}-n-1)\omega_4 \right) S_2\left( i\sigma_{a,\hat{a}} - (\hat{n}-n)\omega_4 + (Y_{a,n}^T - \hat{m})\omega_3 \right)}{S_2\left( i\sigma_{\hat{a},a} - (Y_{a,n}^T - \hat{m} + 1)\omega_3 + (\hat{n}-n)\omega_4 \right) S_2\left( i\sigma_{a,\hat{a}} - (\hat{n}-n+1)\omega_4 + (Y_{a,n}^T - \hat{m})\omega_3 \right)} \right)
$$

$$
\prod_{m=1}^{\hat{m}-1} \frac{S_2\left( -(\hat{m}-m)\omega_3 + (Y_{\hat{a},m} - \hat{n})\omega_4 \right) S_2\left( -(Y_{\hat{a},m} - \hat{n} + 1)\omega_4 + (\hat{m}-m-1)\omega_3 \right)}{S_2\left( -(\hat{m}-m+1)\omega_3 + (Y_{\hat{a},m} - \hat{n})\omega_4 \right) S_2\left( -(Y_{\hat{a},m} - \hat{n} + 1)\omega_4 + (\hat{m}-m)\omega_3 \right)}
$$

$$
\prod_{n=1}^{\hat{n}-1} \frac{S_2\left( -(Y_{\hat{a},n} - \hat{m} + 1)\omega_3 + (\hat{n}-n-1)\omega_4 \right) S_2\left( -(\hat{n}-n)\omega_4 + (Y_{\hat{a},n} - \hat{m})\omega_3 \right)}{S_2\left( -(Y_{\hat{a},n} - \hat{m} + 1)\omega_3 + (\hat{n}-n)\omega_4 \right) S_2\left( -(\hat{n}-n+1)\omega_4 + (Y_{\hat{a},n} - \hat{m})\omega_3 \right)} \frac{1}{S_2(-\omega_3) S_2(-\omega_4)}.
$$

The first factor comes from $i = \hat{a}$ and all contributions are from column $\hat{n}$. The second factor is from $j = \hat{a}$ and all contributions come from row $\hat{m}$. In the last two lines, where $i = j = \hat{a}$, contributions come both from column $\hat{n}$ and row $\hat{m}$, and the box at $(\hat{n}, \hat{m})$. $Y_{b,m}$ denotes the length of the $m$-th row in the diagram $Y_b$, and similar $Y_{b,n}^T$ is the height of the $n$-th column in $Y_b$. The ranges of $a$ and $b$ are the same, so we can just use one multiplication. We also note that the last two lines are the abelian expression for the diagram $Y_{\hat{a}}$, hence we have

$$
\frac{Z_{\vec{Y}_+}}{Z_{\vec{Y}}} = \frac{Z_{Y_{+\hat{a}}}}{Z_{Y_{\hat{a}}}} \prod_{b \neq \hat{a}}^{N} \frac{1}{S_2(i\sigma_{b,\hat{a}} - \omega_3 + (Y_{b,\hat{m}} - \hat{n})\omega_4) S_2(i\sigma_{\hat{a},b} - (Y_{b,\hat{m}} - \hat{n} + 1)\omega_4)} \tag{B.28}
$$

$$
\prod_{m=1}^{\hat{m}-1} \frac{S_2\left( i\sigma_{b,\hat{a}} - (\hat{m}-m)\omega_3 + (Y_{b,m} - \hat{n})\omega_4 \right) S_2\left( i\sigma_{\hat{a},b} - (Y_{b,m} - \hat{n} + 1)\omega_4 + (\hat{m}-m-1)\omega_3 \right)}{S_2 h\left( i\sigma_{b,\hat{a}} - (\hat{m}-m+1)\omega_3 + (Y_{b,m} - \hat{n})\omega_4 \right) S_2\left( i\sigma_{\hat{a},b} - (Y_{b,m} - \hat{n} + 1)\omega_4 + (\hat{m}-m)\omega_3 \right)}
$$

$$
\prod_{n=1}^{Y_{b,\hat{m}}} \frac{S_2\left( i\sigma_{\hat{a},b} - (Y_{b,n}^T - \hat{m} + 1)\omega_3 + (\hat{n}-n-1)\omega_4 \right) S_2\left( i\sigma_{b,\hat{a}} - (\hat{n}-n)\omega_4 + (Y_{b,n}^T - \hat{m})\omega_3 \right)}{S_2\left( i\sigma_{\hat{a},b} - (Y_{b,n}^T - \hat{m} + 1)\omega_3 + (\hat{n}-n)\omega_4 \right) S_2\left( i\sigma_{b,\hat{a}} - (\hat{n}-n+1)\omega_4 + (Y_{b,n}^T - \hat{m})\omega_3 \right)}.
$$

As for the abelian case, many of the factors actually cancel against each other. We can write the expression in terms of boxes $A_{b,i}$ which can be added to the diagram $Y_b$ as well as the right lower boxes $B_{b,i}$ of the rectangles making up the diagram $Y_b$. Tallying up the factors, we end

up with the following expression

$$
\frac{Z_{\vec{Y}_+}}{Z_{\vec{Y}}} = \frac{1}{S_2(-\omega_3)S_2(-\omega_4)} \frac{\prod_i S_2(B_{\hat{a},i} - A_{\hat{a},x})S_2(A_{\hat{a},x} - B_{\hat{a},i} - \omega_3 - \omega_4)}{\prod_{i \neq x} S_2(A_{\hat{a},i} - A_{\hat{a},x} - \omega_3 - \omega_4)S_2(A_{\hat{a},x} - A_{\hat{a},i})}
$$
$$
\prod_{b \neq \hat{a}}^{N} \frac{\prod_i S_2(i\sigma_{b,\hat{a}} + B_{b,i} - A_{\hat{a},x})S_2(i\sigma_{\hat{a},b} + A_{\hat{a},x} - B_{b,i} - \omega_3 - \omega_4)}{\prod_i S_2(i\sigma_{b,\hat{a}} + A_{b,i} - A_{\hat{a},x} - \omega_3 - \omega_4)S_2(i\sigma_{\hat{a},b} + A_{\hat{a},x} - A_{b,i})},
$$
(B.29)

where $A_{\hat{a},x}$ is the position of the box that takes us from $\vec{Y}$ to $\vec{Y}_+$.

Now let us compare (B.29) against the ADHM integral

$$
\frac{Z_{\vec{Y}_+}}{Z_{\vec{Y}}} = \frac{S_2(-\omega_3 - \omega_4)}{S_2(-\omega_3)S_2(-\omega_4)} \int \frac{d\phi_{k+1}}{i\sqrt{\omega_1\omega_2}} \prod_{a=1}^{N} \frac{1}{S_2(\phi_{k+1} - a_a)S_2(-\phi_{k+1} + a_a - \omega_3 - \omega_4)}
$$
$$
\times \prod_{i=1}^{k} \prod_{\eta=\pm 1} \frac{S_2(\eta(\phi_{k+1} - \hat{\phi}_i))S_2(\eta(\phi_{k+1} - \hat{\phi}_i) - \omega_3 - \omega_4)}{S_2(\eta(\phi_{k+1} - \hat{\phi}_i) - \omega_3)S_2(\eta(\phi_{k+1} - \hat{\phi}_i) - \omega_4)},
$$
(B.30)

where $\hat{\phi}_i$, $i = 1, \dots, k$, is the pole picked up by the contour integration of $\phi_i$ according to the Young diagrams $\vec{Y}$. Explicitly they are at $a_b + (m-1)\omega_3 + (n-1)\omega_4$ for box $(n,m)$ in diagram $Y_b$, $b = 1, \dots, N$. Then, we can convince ourselves once again by considering the diagrams that most factors cancel, leaving us with the integral

$$
\frac{Z_{Y_+}}{Z_Y} = \frac{S_2(-\omega_3 - \omega_4)}{S_2(-\omega_3)S_2(-\omega_4)} \int \frac{d\phi_{k+1}}{2\pi i}
$$
$$
\times \prod_{a=1}^{N} \frac{\prod_i S_2(\phi_{k+1} - a_a - B_{a,i} - \omega_3 - \omega_4)S_2(a_a - \phi_{k+1} + B_{a,i})}{\prod_i S_2(\phi_{k+1} - A_{a,i} - a_a)S_2(a_a - \phi_{k+1} + A_{a,i} - \omega_3 - \omega_4)}.
$$
(B.31)

The different poles that we now can pick up (or want to allow to be picked up) are at $\phi_{k+1} = A_{a,i} + a_a$. If we do this for $a = \hat{a}$ and $i = x$ and set $a_a - a_b = i\sigma_{ab}$, we find eq. (B.29), confirming our conjecture.

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
