# Peer review of "Seven-dimensional super Yang-Mills at negative coupling"

_SciPost Physics, doi:SciPost Phys. 14, 028 (2023)_

## Round 2 · Referee Report · Anonymous · 2022-8-23

Report
This paper studies the 7d SYM on the round and squashed $S^7$ using localization technique and obtains exact formulae for partition function. In particular, it discusses in detail the theory for negative value of YM coupling squared. Naively the contribution of instanton gets enhanced in such a phase, but it is shown that the enhancement is canceled by Nekrasov partition function. This paper also gives some evicences that the theory at negative coupling-squared can be described by a 7d supergravity, which definitely deserves further study.
Some detail regarding the computation of membrane instanton contribution is presented in the appendix.
Overall the paper is very interesting and clearly written, so I recommend it for publication in SciPost.
Requested changes
Here are some (possible) typos and suggestions.
1- (page 6) just after equation (2.5) there is an equation "$\imath_vF=F_V$" which does not make sense.
2- (page 15) It will be nice to explain the condition or the limit in which the weak equality "$\approx$" in equation (3.10) holds. Also, the equality "$=$" in equation (3.12) should be replaced with "$\approx$"
3- (page 15) In the equation right after (3.13), $q=e^{-\frac{2\pi\delta\sigma}{\omega_2}}$, replace $\omega_2$ by $\omega_1$
4- (page 28) In equation (B.20), the product in the denominator of RHS is over $i=0,\cdots,n_r$ such that $j\ne x$. It should be $i\ne x$.
5- (page 29) The range of integer variables $n,m$ in the Figure 2(a) should be the same as those in (B.19)

---

## Round 2 · Referee Report · Anonymous · 2022-9-23

Strengths
1-Uses useful approximations to give significant evidence to the physically important conjecture that the seven-dimensional super Yang-Mills theory has a parameter region where the low-energy theory is a supergravity theory.
2-The cancellation of poles at the one-instanton level is highly non-trivial and demonstrates the consistency of the results in the paper.
Weaknesses
1-The analysis is based on an approximate solution to the saddle point equation (2.13) of a matrix integral.
2-The cancellation of poles is done only at the one-instanton level. It is desirable to do the same for higher instanton numbers.
Report
The work under review performs the supersymmetric localization computation of the partition function of the seven-dimensional super Yang-Mills theory with 16 supercharges on a squashed seven-sphere $\mathbb{S}^7$. The authors apply useful approximations to give significant evidence to the physically important conjecture that the seven-dimensional super Yang-Mills theory has a parameter region where the low-energy theory is a supergravity theory. They demonstrate the cancellation of poles at the one-instanton level, which is highly non-trivial and demonstrates the consistency of the results in the paper.
The results and the conclusions of the paper are interesting and important.
The paper is written clearly, and provide adequate references in the literature.
I recommend the paper for publication in the journal.
Requested changes
1-In the manipulation in (2.10), $\sum_i \sigma_i$ is set to zero. So the authors seem to considering $SU(N)$ as the gauge group. On the other hand the measure factor $\prod_{i=1}^N {\rm d}\sigma_i$ in (2.9) would be appropriate for $U(N)$ but not for $SU(N)$. Some clarifying comments seem required.
2-Above (3.9), they write that "the two eigenvalues are widely separated". Because the gauge group is $SU(2)$, actually there is only one eigenvalue $\sigma_{12}$. $\sigma_1$ and $\sigma_2$ do not exist independently.
3-Below (2.25), "equals equals" should be "equals''.
4-Below (2.31), "expressed" should be "suppressed".
5-Above (3.1) "Localizing" should be "localizing".
6-Above (4.1), "given given" should be "given".

---

## Round 3 · Author Response

We thank the referees for their careful reading of the manuscript. We have included all of their suggestions in the new manuscript, a list of all the changes is given.

---

## Round 3 · List of Changes

1. Before eq. (2.6) added \kappa\wedge to \iota_v F
    1. Before eq. (2.8) “\hat F^+ decompose” changed to “\hat F^+ decomposes”
    2. eq. (2.9) and (2.11) added delta functions to make it SU(N) partition functions
    3. after (2.25) deleted one “equals”
    4. after (2.31) changed “expressed” to “suppressed”
    5. before (3.1) changed “Localizing” to “localizing”
    6. after (3.1) changed “i_v F” to “\iota_v F”
    7. after (3.2) added the definition of \varrho_5
    8. before (3.9) changed “the two eigenvalues are widely separated” to “the eigenvalue $\sigma_1=-\sigma_2$ is large”
    9. after (3.10) changed “where the last step […]” to “where we have again expanded around the saddle point $\sigma_{12}=-\frac{4\pi^2\mathcal{R}}{g_{YM}^2}$ in the small negative $g_{YM}^2$ limit. The last step [...]”
    10. in (3.12) changed = to \approx
    11. after (3.13) in the definition of q changed \omega_2 to \omega_1
    12. before (4.1) removed one “given”
    13. In (4.1) changed k to \kappa
    14. before (4.14) added the article to “the dominant contribution”
    15. in (B.20) changed “j\neq x” to “i\neq x”
    16. Figure 2(a) exchanged “\hat{m}-1” and “\hat{n}-1”

---

## Editorial Decision

published